# AllClear: A Comprehensive Dataset and Benchmark for Cloud Removal in Satellite Imagery

**Hangyu Zhou**[1] [*], **Chia-Hsiang Kao**[1][*], **Cheng Perng Phoo**[1],
**Utkarsh Mall**[2], **Bharath Hariharan**[1], **Kavita Bala**[1]

[1]Computer Science, Cornell University
[2]Computer Science, Columbia University

## Abstract

Clouds in satellite imagery pose a significant challenge for downstream applications. A major challenge in current cloud removal research is the absence of a comprehensive benchmark and a sufficiently large and diverse training dataset. To address this problem, we introduce the largest public dataset — *AllClear* for cloud removal, featuring 23,742 globally distributed regions of interest (ROIs) with diverse land-use patterns, comprising 4 million images in total. Each ROI includes complete temporal captures from the year 2022, with (1) multi-spectral optical imagery from Sentinel-2 and Landsat 8/9, (2) synthetic aperture radar (SAR) imagery from Sentinel-1, and (3) auxiliary remote sensing products such as cloud masks and land cover maps. We validate the effectiveness of our dataset by benchmarking performance, demonstrating the scaling law — the PSNR rises from 28.47 to 33.87 with $30\times$ more data, and conducting ablation studies on the temporal length and the importance of individual modalities. This dataset aims to provide comprehensive coverage of the Earth's surface and promote better cloud removal results.

## 1 Introduction

Satellite image recognition enables environmental monitoring, disaster response, urban planning [Pham et al., 2011, Wellmann et al., 2020], crop-yield prediction [Doraiswamy et al., 2003], and many more applications, but is held back significantly due to occlusion by clouds. Roughly 67% of the Earth's surface is covered by clouds at any given moment [King et al., 2013]. The limited availability of cloud-free captures is especially problematic for time-sensitive events like wildfire control [Kyzirakos et al., 2014, Thangavel et al., 2023] and flood damage assessment [Rahman and Di, 2020]. Consequently, developing effective cloud removal techniques is crucial for maximizing the utility of remote sensing data in various domains.

A major challenge holding back research into cloud removal is the lack of comprehensive datasets and benchmarks. A survey of publicly available datasets for cloud removal (Table 1) reveals several problems. First, most existing datasets are sampled from a small set of locations and thus have limited geographical diversity [Ebel et al., 2020, Huang and Wu, 2022, Ebel et al., 2022], impacting both the effectiveness of training and the rigor of evaluation. Second, many existing datasets filter out very cloudy images (e.g., more than 30% cloud coverage), thus preventing trained models from tackling practical situations with extensive cloud cover [Sarukkai et al., 2020, Requena-Mesa et al., 2021] (Figure 1). Third, some existing benchmarks use ground-truth cloud-free images captured at a very different time point from the time the input images are captured [Sarukkai et al., 2020, Ebel et al., 2022]. This means that many changes may have occurred on the ground between the

---

[*]Contributed Equally. Correspondence to Hangyu Zhou and Chia-Hsiang Kao at {hz477,ck696}@cornell.edu

| Dataset | Regions | # ROIs | # Images | Satellites |
|---|---|---|---|---|
| STGAN [Sarukkai et al., 2020] | Worldwide | 945 | 3,101 | Sentinel-2 |
| Sen2_MTC [Huang and Wu, 2022] | Worldwide | 50 | 13,669 | Sentinel-2 |
| EarthNet2021 [Requena-Mesa et al., 2021] | Europe | 32,000 | 960,000 | Sentinel-2 |
| SEN12MS-CR [Ebel et al., 2020] | Worldwide | 169 | 366,654 | Sentinel-1/2 |
| SEN12MS-CR-TS [Ebel et al., 2022] | Worldwide | 53 | 917,580 | Sentinel-1/2 |
| AllClear | Worldwide | 23,742 | 4,354,652 | Sentinel-1/2, Landsat-8/9 |

**Table 1:** Summary of publicly available cloud removal datasets.

capture of the input and the target images, introducing noise in the evaluation. Finally, existing datasets incorporate a very limited set of sensors/modalities (i.e., Sentinel-2), limiting the information available to models for faithful cloud removal.

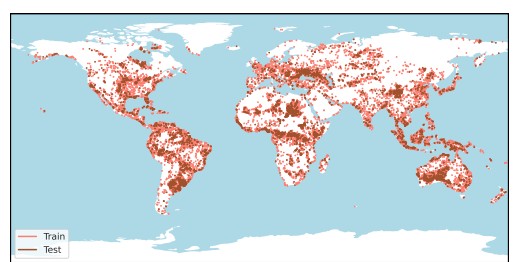 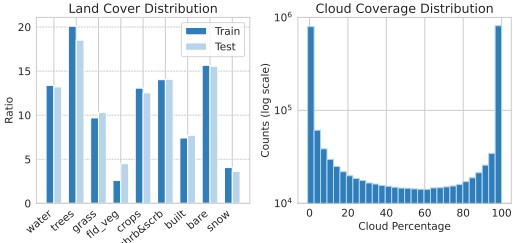

**Figure 1:** Left: Geographical distribution of *AllClear* ROIs; middle: land cover distribution of *AllClear* for training and testing set; right: cloud coverage distribution of the entire *AllClear* dataset.

To address these limitations and facilitate future research in cloud removal, we introduce the largest and most comprehensive dataset to date, *AllClear*. To ensure sufficient coverage of the planet's diversity, *AllClear* includes 23,742 regions of interest (ROIs) scattered across the globe with diverse land cover patterns, resulting in four million multi-spectral images. *AllClear* includes data from three different satellites (i.e., Sentinel-1, Sentinel-2, and Landsat-8/9) captured over a year (2022) at each ROI, allowing models to better interpolate missing information. We use this dataset to create a more rigorous sequence-to-point benchmark with more temporally aligned ground truth. Finally, besides the enormous amount of raw satellite images, we also curated a rich set of metadata for each individual image (e.g., geolocation, timestamp, land cover map, cloud masks, etc.) to support building future models for the cloud removal challenge as well as to enable stratified evaluation.

We evaluate existing state-of-the-art on AllClear and find that existing models are undertrained; training on our larger and more diverse training set significantly improves performance. We also find that models that use the full suite of available sensors as well as a longer temporal sequence of captures perform much better. Taken together, our contributions are:

- We introduce to-date the largest dataset for cloud removal, as well as a comprehensive and stratified evaluation benchmark,

- We demonstrate that our significantly larger and more diverse training set improves model performance, and

- We show empirically the importance of leveraging multiple sensors and longer time spans.

## 2 Background

### 2.1 Existing Cloud Removal Datasets

Advances in cloud removal research for satellite imagery have led to the development of several datasets with unique characteristics and limitations. STGAN introduced two cloud removal datasets and established the multi-temporal task format of using three images as input [Sarukkai et al., 2020]. However, the dataset discards all image crops with more than 30% cloud cover, leading to only

3K images. Following STGAN, Huang and Wu [2022] find that the annotations in STGAN can be incorrect and propose Sen2_MTC with four times more images. The Sen2_MTC dataset first samples 50 tiles globally and proceeds to divide the large tile into pieces, restricting the sampling regional diversity. STGAN and Sen2_MTC also do not describe their *data processing levels* (e.g., level-1C Top-of-Atmosphere or level-2A Surface Reflectance imagery), making it hard to compare models trained on different datasets. Different from the STGAN and Sen2_MTC datasets, the SEN12MS-CR dataset features synthetic-aperture radar (SAR) images to augment the optics imagery. However, it has a single image pair per data point. The successor is SEN12MS-CR-TS [Ebel et al., 2022], featuring multi-temporal (multiple images per location) multi-modality paired images. For each location, 30 Sentinel-1 and Sentinel-2 images from 2018 are temporally aligned and paired as spatiotemporal patches. However, the temporal differences between the two modalities can be as large as 14 days, and the temporal difference between the input and the target can be as large as a year, resulting in noise in the evaluation. In addition, the authors construct a sequence-to-point cloud removal dataset where images with more than 50% cloud coverage are excluded. EarthNet2021 [Requena-Mesa et al., 2021] also provides sequences of carefully curated Sentinel-2 images with a spatial resolution of 20m and bands of RGB and Infrared. However, the dataset excluded spatiotemporal patches with high cloud coverage and is thus not an ideal dataset for cloud removal.

## 2.2 Cloud Removal Methodology

Early work on cloud removal used a conditional GAN to map a single image to its cloudless version conditioning on the NIR channel [Enomoto et al., 2017] or SAR images [Grohnfeldt et al., 2018]. These early attempts fall short of generalizing to real cloudy images [Ebel et al., 2020, Stucker et al., 2023]. Singh and Komodakis [2018] and Ebel et al. [2020] improve this setup by using a cycle-consistency loss. Other approaches learn the mapping from SAR images to their corresponding multi-spectral bands [Bermudez et al., 2018, 2019, Wang et al., 2019, Fuentes Reyes et al., 2019]. More recently, with the rise of transformers, multi-head attention modules have been introduced for cloud removal tasks. Yu et al. [2022] cast the cloud as image distortion and designs a distortion-aware module to restore the cloud-free images. Zou et al. [2023a] utilize multi-temporal inputs along with a multi-scale attention autoencoder to exploit the global and local context for reconstruction. Ebel et al. [2023] adopt a multi-temporal inputs and attention autoencoder and estimate the aleatoric uncertainty of the prediction, which controls the quality of the reconstruction for risk-mitigation applications. Jing et al. [2023] and Zou et al. [2023b] propose to utilize diffusion training objective for cloud-free image generation where the inputs only rely on the optimal images and SAR imagery is not taken into consideration. Similarly but more generally, Khanna et al. [2023] propose a generative foundation model for satellite imagery, but is not tailored for the cloud removal task.

# 3 Dataset

## 3.1 Regions-of-Interest Selection

We choose our ROIs to satisfy two objectives: (a) coverage of most of the land surface and (b) a balanced sampling of land cover types. This balanced sampling in particular ensures that smaller but more popular locations like cities are as well represented as the large swathes of wilderness. To get these ROIs, we follow a two-step procedure: curating a pool of ROI candidates and then building train/benchmark subgroups balanced across land cover types, as shown in Figure 1. This ensures both the benchmark and the training sets contain a sufficient amount of data representing various land cover types.

For curating the ROI pool, unlike previous work that followed random ROI selection [Sarukkai et al., 2020, Huang and Wu, 2022, Ebel et al., 2020, 2022, Xu et al., 2023], we use grid sampling to select an ROI every $0.1°$ latitude and every $0.1°\cos(\theta)$ longitude, where $\theta$ is the latitude, from 90°S to 90°N. The intuition behind this approach is that the same $0.1°$ longitude can represent 11.1 km at the equator and 4.35 km at 67° latitude. This weighting provides a simple yet effective method for not over-sampling high-latitude areas. By excluding ocean areas using the `GeoPandas` package, we select a total of 1,087,947 ROIs.

Next, we select ROIs from the pool to achieve a more balanced dataset over land-cover use while considering the natural imbalance of land cover distribution on the earth's surface. We leverage the land cover data from the Dynamic World product [Brown et al., 2022] from Google Earth Engine,

which is a 10-meter resolution Land Use / Land Cover (LULC) dataset containing class probabilities and label information for nine classes: water, tree, grass, flooded vegetation, crops, shrub and scrub, built, bare, and snow and ice. Specifically, we calculate the all-year median of the LULC in 2022 as an estimate for the land use and land cover for each ROI. We iteratively select ROIs from the candidate pool such that the average land cover for all classes (except snow and ice) is greater than 10 percent in the benchmark set and 5 percent in the train set.

Finally, for a fairer comparison with models trained on previous datasets, we take an additional measure to exclude the ROIs that are close to the SEN12MS-CR-TS dataset [Ebel et al., 2022]. Specifically, the size of tiles in the SEN12MS-TR-CS dataset is $40 \times 40$ km$^2$. So we exclude the ROIs in AllClear that are within a 50 km radius of the ROIs in SEN12MS-CR-TS.

## 3.2 Data Preparation

*AllClear* contains three different types of open-access satellite imagery made available by the Google Earth Engine (GEE) platform [Gorelick et al., 2017]: Sentinel-2A/B [Drusch et al., 2012], Sentinel-1A/B [Torres et al., 2012], and Landsat 8/9 [Williams et al., 2006]. For Sentinel-2, we collected all thirteen bands of Level-1C orthorectified top-of-atmosphere (TOA) reflectance product. For Sentinel-1, we acquired the S1 Ground Range Detected (GRD) product with two polarization channels (VV and VH). As for Landsat 8/9, we collected all twelve bands of Collection 2 Tier 1 calibrated TOA reflectance product. All the raw images in *AllClear* were resampled to 10-meter resolution. We follow the default GEE preprocessing steps during all the downloading process. In addition, we include the Dynamic World Land Cover Map for all the Sentinel-2 imagery [Brown et al., 2022]. For each selected ROI, our goal is to collect all $2.56 \times 2.56$ km$^2$ patches in 2022 with a spatial resolution of 10 meters. We adopt the Universal Transverse Mercator (UTM) coordinate reference system (CRS), following Ebel et al. [2020, 2022], Zhao et al. [2023], which divides the Earth into 60 zones, each spanning 6 degrees of longitude, to ensure minimal distortion, especially along the longitude axis. Since satellite imagery do not necessarily conform to the boundaries of UTM zones, gaps (NaN values) can occur where the tile data does not cover the entire ROI. In such cases, we exclude all images containing NaN values to maintain data quality.

**Data Preprocessing.** For Sentinel-1, following Ebel et al. [2022], we clip the values in the VV channel of S1 to $[-25; 0]$ and those of the VH channels to $[-32.5, 0]$. For Sentinel-2 and Landsat 8/9, we clip the raw values to $[0, 10000]$ [Ebel et al., 2022, Huang and Wu, 2022]. The values are then normalized to the range of $[0, 1]$.

**Cloud and Shadow Mask Computation.** The cloud and shadow masks are indispensable to this dataset as they are used for guiding evaluation metric computation by masking out regions where there are clouds and shadows in the target images. To obtain the cloud mask, we use the S2 Cloud Probability dataset available on Google Earth Engine. This dataset is built by using S2cloudless [Zupanc, 2017], an automated cloud-detection algorithm for Sentinel-2 imagery based on a gradient boosting algorithm, which shows the best overall cloud detection accuracy on opaque clouds and semi-transparent clouds in the Hollstein reference dataset [Hollstein et al., 2016, Skakun et al., 2022] and the LCD PixBox dataset [Paperin et al., 2021, Skakun et al., 2022].

As for the shadow mask, ideally the cloud shadows can be estimated using the sun azimuth and cloud height but the latter information cannot be obtained. We therefore proceed with curating the shadow mask following documentation in Google Earth Engine [jdbcode, 2023]. The shadow is estimated by computing dark pixels and projecting cloud regions. For the dark pixels, we use the Scene Classification Map (SCL) band values from Sentinel-2 to remove water pixels, as water pixels can resemble shadows. We then threshold the NIR pixel values with a threshold of 1e-4 to create a map of dark pixels. Finally, we take the intersection of the dark pixel map and the projected cloud regions to obtain the cloud shadow masks.

## 3.3 Benchmarking Task Setup and Evaluation

For evaluation, we construct a sequence-to-point task using our AllClear dataset with train, validation, and test splits of 278,613, 14,215, and 55,317 samples, respectively. Each instance contains three input images $(u_1, u_2, u_3)$, a target clear image $(v)$, input cloud and shadow masks, target cloud and shadow masks, timestamps, and metadata such as latitude, longitude, sun elevation angle, and sun azimuth. Sentinel-2 images are considered the main sensor modality, while sensors such as Sentinel-1

and Landsat-8/9 are auxiliary. Unlike previous datasets [Sarukkai et al., 2020, Requena-Mesa et al., 2021, Ebel et al., 2022], we do not threshold the cloud coverage in the input images. We also provide multiple options for cloud and shadow masks with different thresholds for users to use.

We address two temporal misalignment problems found in previous datasets: misalignment between source and target images (where the difference can be months apart) and misalignment when pairing main sensors with auxiliary sensors (where the difference can be at most two weeks) [Ebel et al., 2022]. To avoid temporal misalignment issues, the target clear images are chosen from four consecutive spatial-temporal patches. In particular, the time stamps of the input and target images are either in the order $[u_1, v, u_2, u_3]$ or in the order $[u_1, u_2, v, u_3]$. This ensures that the target image does not include any novel or unseen changes that occurred after the capture of the cloudy images. For auxiliary sensors, we select the auxiliary satellite images within a two-day difference from the respective Sentinel-2 images. We fill the corresponding channels with ones if no auxiliary sensor images match are available. More details about the construction of these inputs and targets is in the supplementary.

Note that our target images may still have some clouds (since it is difficult to get a cloud-free image within each time span). To reach a balance between having diverse scenarios and limit metric inaccuracy, we set target images to have less than 10% cloud and shadow (combined) coverage and exclude the cloudy pixels when calculating the metrics. We modified various pixel-based metrics to compute only over the cloud-free areas. We adopt the following metrics common in cloud removal literature: mean absolute error (MAE) [Hodson, 2022], root mean square error (RMSE) [Hodson, 2022], peak signal-to-noise ratio (PSNR) [Hore and Ziou, 2010], spectral angle mapper (SAM) [Kruse et al., 1993], and structural similarity index measure (SSIM) [Wang et al., 2004].

## 4 Experiments

We next evaluate the usefulness of our dataset for both evaluation and training.

### 4.1 Benchmarking prior methods on the AllClear test set

**Selection of SoTA model architecture.** We choose the state-of-the-art pre-trained models UnCRtainTS [Ebel et al., 2023], U-TILISE [Stucker et al., 2023], CTGAN [Huang and Wu, 2022], PMAA [Zou et al., 2023a], and DiffCR [Zou et al., 2023b] to benchmark on our AllClear dataset. For the evaluation, all models receive three images as input. Specifically, they receive both Sentinel-2 and Sentinel-1 images concatenated along the channel dimension.

**Simple baselines** To better contextualize model performance, we follow previous works [Ebel et al., 2022, 2023] and include two simple baselines: "Least Cloudy" and "Mosaicing". The former simply uses the input image with the least cloud and shadow coverage as the output. "Mosaicing" operates in the following way: for each image coordinates in the input images if only one image is clear, we directly copy its pixel value; if more than one clear images exist, we take the average of these clear pixel values; if there is no clear image, we fill the gap with 0.5.

**Results.** The quantitative and qualitative results are shown in Table 2 and Figure 8, respectively. We first notice that simple baselines *least cloudy* and *mosaicing* perform well on the dataset. UnCRtainTS performs slightly better than these simple baselines in terms of SSIM and SAM. On the other hand, the U-TILISE model falls short of reaching the performance of the simple baselines. Since U-TILISE is a sequence-to-sequence model, we adopt it for sequence-to-point evaluation by choosing the image from the output sequence with the lowest MAE score as the model output. Notably, the training of U-TILISE involves adding sampled cloud masks to the cloud-free images as inputs, and it is trained to recover the original cloud-free sequence. The model is evaluated in a similar manner. The distribution disparity between the sampled cloud masks and the real clouds may contribute to the low score of U-TILISE in the real scenario. For the good performance of *least cloudy* and *mosaicing*, we conjecture that this is because of the small temporal gap in AllClear between input and target images, so simply averaging or choosing from the input images is likely to yield good results.

Notably, models pre-trained on STGAN and Sen2_MTC datasets (specifically CTGAN, PMAA, and DiffCR) performed below simple baselines. Due to insufficient documentation of imagery specifications and pre-processing protocols in these datasets, we excluded these pre-trained models from subsequent analysis.

**Table 2:** Benchmark performance of previous SoTA models evaluated on our AllClear benchmark dataset. The best performing values are in **bold** and the second best is underlined.

| Model | Training Dataset | PSNR (↑) | SSIM (↑) | SAM (↓) | MAE (↓) |
|---|---|---|---|---|---|
| Least Cloudy | - | 28.864 | 0.836 | 6.982 | 0.078 |
| Mosaicing | - | **29.824** | 0.754 | 23.58 | 0.045 |
| UnCRtainTS [Ebel et al., 2023] | SEN12MS-CR-TS | 29.009 | **0.898** | **5.972** | **0.039** |
| U-TILISE [Stucker et al., 2023] | SEN12MS-CR-TS | 24.660 | 0.807 | 7.765 | 0.083 |
| CTGAN [Huang and Wu, 2022] | Sen2_MTC | 27.783 | 0.840 | 8.800 | 0.041 |
| PMAA [Zou et al., 2023a] | STGAN | 12.455 | 0.460 | 8.072 | 0.240 |
| | Sen2_MTC | 24.328 | 0.768 | 8.680 | 0.078 |
| DiffCR [Zou et al., 2023b] | STGAN | 17.998 | 0.642 | 9.512 | 0.117 |
| | Sen2_MTC | 25.220 | 0.744 | 9.382 | 0.060 |

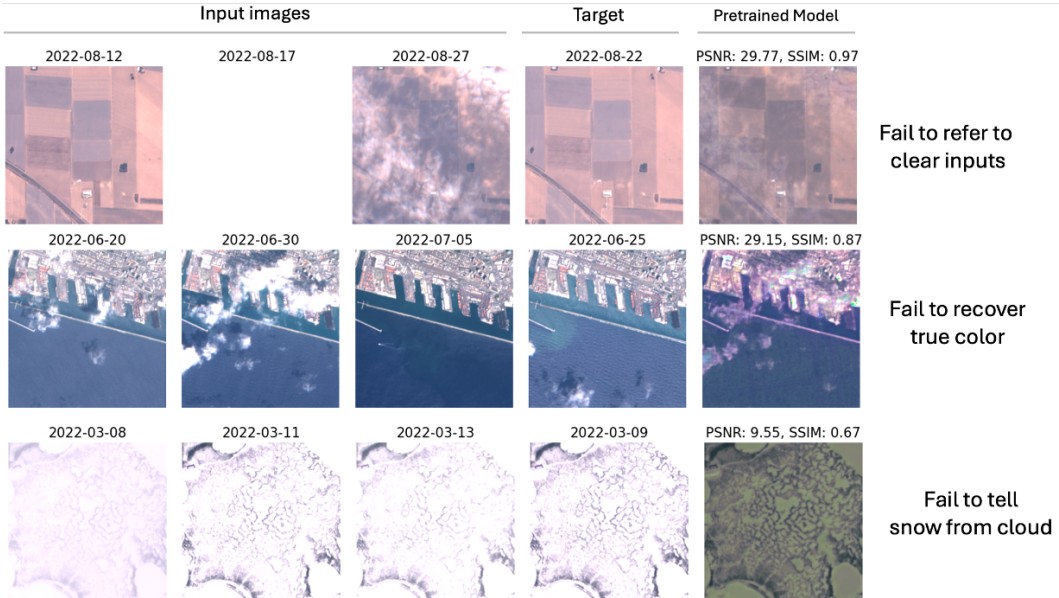

**Figure 2:** Failure case from UnCRtainTS [Ebel et al., 2023], a previous SOTA model trained on the SEN12MS-CR-TS [Ebel et al., 2022] cloud removal dataset.

**Failure cases.** To understand the performance of the state-of-the-art better, we visualize the output images generated using the state-of-the-art model UnCRtainTS [Ebel et al., 2023], which was trained on the SEN12MS-CR-TS dataset [Ebel et al., 2022]. In Figure 2, we evaluate the pre-trained model on AllClear testing cases where it receives three cloudy images as input. Overall, we observe three primary failure modes in the model's performance: (1) The model fails to draw from clear input images, particularly when the other two images are cloudy. This issue may arise because the model was trained exclusively on images with less than 50% cloud coverage, as noted by the authors [Ebel et al., 2023]. (2) The model often struggles to recover the correct color spectrum, even when the input images are mostly clear. We hypothesize that this is due to the relatively small dataset size, leading to a lack of generalization ability. (3) The model frequently fails to generalize to snow-covered land. We speculate that this is due to insufficient sampling of diverse snowy regions during training.

## 4.2 Training on AllClear

We next evaluate the benefits of training on AllClear. For this purpose, we use UnCRtainTS given its good performance on prior benchmarks [Ebel et al., 2023]. To evaluate if there is any domain difference between AllClear and the previous SEN12MS-TR-CS dataset, we first run an equal-training-set-size comparison. We train UnCRtainTS on a *subset* of AllClear that is of the same size as the training set size used in UnCRtainTS training, which is 10,167 data points. We also follow the training hyperparameters as in the original paper to avoid extra tuning. As shown in Table 3,

**Table 3:** Benchmark Performance for UnCRtainTs models retrained on AllClear.

| Evaluation Dataset | Training Dataset (fraction used) | PSNR (↑) | SSIM (↑) | SAM (↓) | MAE (↓) |
|---|---|---|---|---|---|
| SEN12MS-CR-TS | SEN12MS-CR-TS | **27.838** | **0.866** | **9.455** | **0.036** |
| | AllClear (3.4%) | 26.256 | 0.847 | 10.411 | 0.041 |
| AllClear | SEN12MS-CR-TS | **29.009** | 0.898 | **5.972** | 0.039 |
| | AllClear (3.4%) | 28.474 | **0.906** | 6.373 | **0.036** |

when both models are evaluated on AllClear (i.e., the bottom two rows in Table 3), we observe that UnCRtainTS models pre-trained on both datasets have comparable results across the four metrics. This suggests that there is no noticeable domain difference between the two datasets.

**Scaling with AllClear.** We next evaluate how much we can scale UnCRtainTS using the large training set available with AllClear. Specifically, we curate a dataset of various scales using random sampling from the training dataset while evaluating on the same validation set. Table 4 shows the results. We find that more training data clearly improves accuracy significantly across all metrics, resulting in a more than 10% improvement in PSNR. Figure 3 shows that with a larger dataset the model is able to better remove clouds and better preserve the color. This suggests that cloud removal models trained on past datasets are in general *undertrained* and AllClear's large training set is useful to help the models fit the data better.

**Table 4:** Scaling law of our model on our AllClear datasets with UnCRtainTS as backbone architecture.

| Fraction of Data | # data point | PSNR (↑) | SSIM (↑) | SAM (↓) | MAE (↓) |
|---|---|---|---|---|---|
| 1% | 2,786 | 27.035 | 0.898 | 5.972 | 0.039 |
| 3.4% | 10,167 | 28.474 | 0.906 | 6.373 | 0.036 |
| 10% | 27,861 | 32.997 | 0.923 | 6.038 | **0.023** |
| 100% | 278,613 | **33.868** | **0.936** | **5.232** | **0.021** |

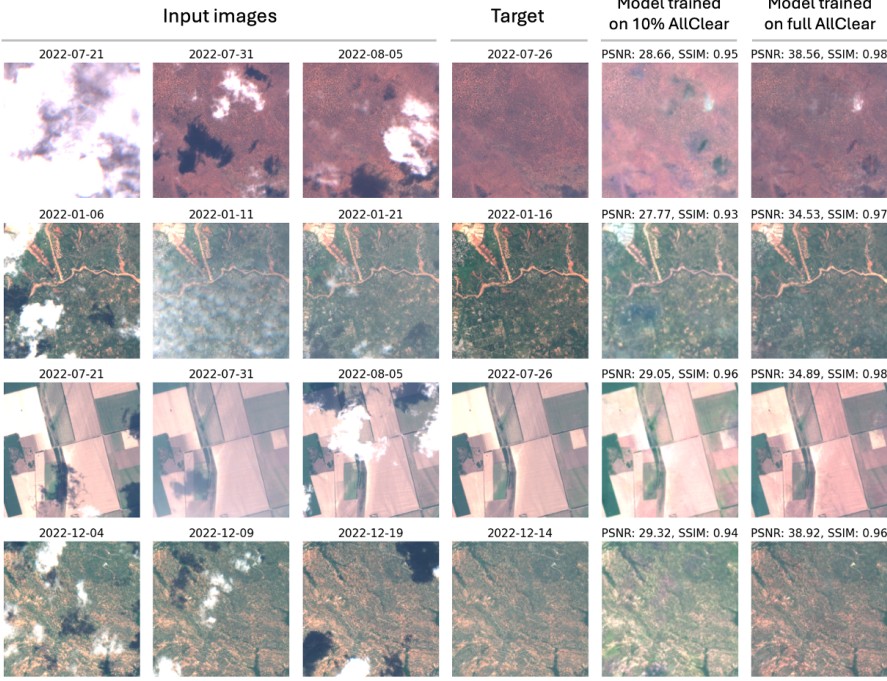

**Figure 3:** Scaling the training dataset by tenfold gives better qualitative results.

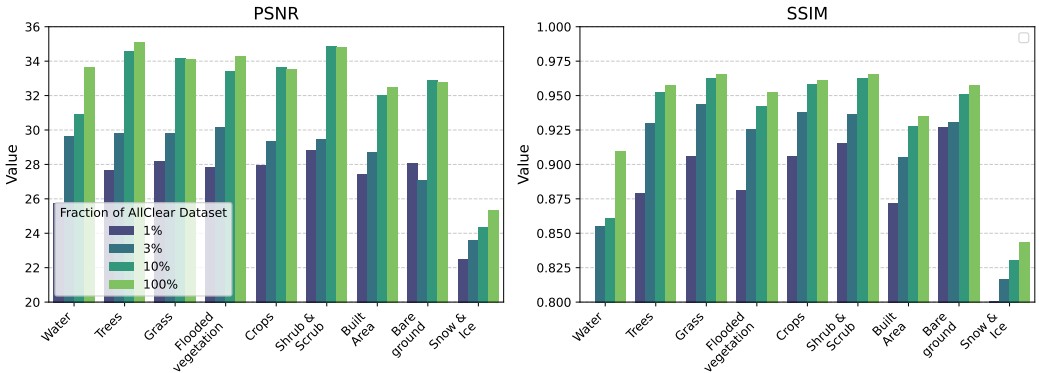

**Figure 4:** Land cover stratified evaluation of models trained with different fractions of the AllClear dataset: 1%, 3.4%, 10%, and 100%.

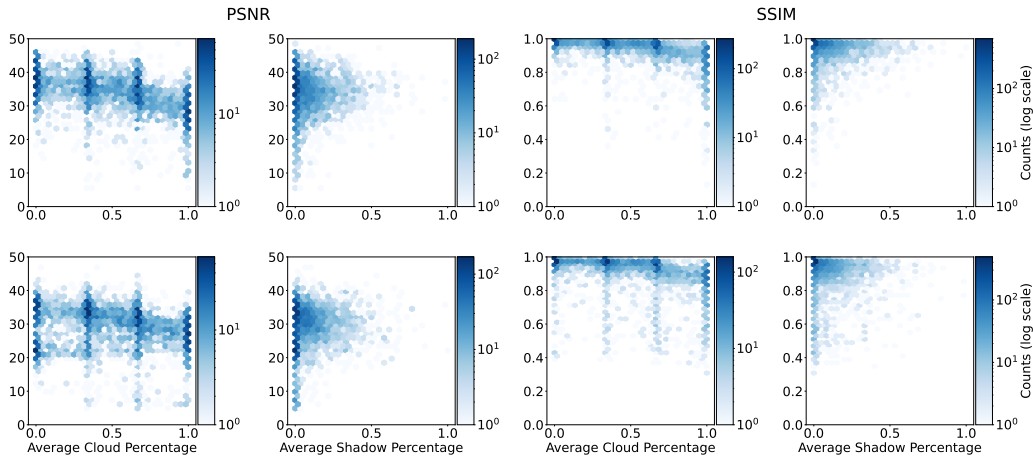

**Figure 5:** Cloud removal quality measured by PSNR (left column) and SSIM (right column) at different cloud and shadow coverage levels. The top row represents models trained on the full AllClear dataset, and the bottom row represents models trained on the SEN12MS-CR-TS dataset.

### 4.3 Stratified evaluations

We use the available land-cover type labels in *AllClear* to conduct a stratified evaluation across land-cover types. As shown in Figure 4, we find that both PSNR and SSIM metrics are generally much worse for both water bodies and snow cover. Water bodies have transient wave patterns, and snow cover is also often transient, which may explain the difficulty of predicting these classes. Snow may also be confused with cloud.

Following past work [Ebel et al., 2022], we also perform a stratified evaluation of accuracy relative to the extent of cloud cover and shadows (Figure 5). For cloud cover, generally performance decreases with cloud percentage, which is expected. Training on a larger dataset (AllClear) substantially improves accuracy for low and medium cloud cover, but not for fully clouded regions. Note that the striped pattern is because of fully cloudy images as explained in the Appendix. Shadows are generally less of a problem, and shadow percentage seems to be uncorrelated with performance.

### 4.4 Effect of various temporal spans

We next use our benchmark to see whether the common practice of using 3 input images is sufficient. We compare two models, one using 3 images and the other using all 12 images captured at that location. Both models are trained on a 10k subset of *AllClear*. The results, shown in Table 5, suggest

that in fact a longer timespan significantly improves accuracy. Future cloud removal techniques should therefore consider longer timespans.

**Table 5:** Effect of different temporal length.

| # Consecutive Frame as Input | PSNR (↑) | SSIM (↑) | SAM (↓) | MAE (↓) |
|---|---|---|---|---|
| 3 | 28.474 | 0.906 | 6.373 | 0.036 |
| 12 | 30.399 | 0.919 | 5.920 | 0.028 |

## 4.5 Ablation study on multi-modality preprocessing strategies

Here we explore the integration of multiple sensors into the input data. As described above, we concatenate multi-spectral Sentinel-2 images with Sentinel-1 and Landsat images to create an input with multiple channels. However, due to the differing revisit intervals of these satellites, there can be gaps in the input sequences, meaning that some Sentinel-2 images may not have corresponding Sentinel-1 or Landsat-8/9 images.

To address these gaps, we experimented with different preprocessing strategies, as shown in Table 6. We discovered that filling the gaps with different constant values significantly impacts the results. Specifically, filling with zeros yielded better performance compared to filling with ones. Also, we provided additional experiments adding an extra input dimension called the "availability mask," which is filled with zeros if there is no paired Sentinel-1 image and ones otherwise, but this approach did not improve results. Additionally, while outcomes regarding using extra Landsat images were inconsistent, filling gaps with zeros for Landsat gave the best results, albeit still lower than using only Sentinel-1 and Sentinel-2 alone. This might be due to the low-resolution of Landsat imagery; we suggest a model redeisgn to fully exploit Landsat images.

**Table 6:** Multi-modality ablation studies. UnCRtainTS models are trained on a 10K subset of samples from our datasets with various setups. *S1* and *LS* denote Sentinel-1 and Landsat images, respectively. Preprocessing methods: *FZ* - Fill zeros, *FO* - Fill ones, *AM* - Availability mask. *FZ/FO* indicates filling gaps with constant zeros/ones when no nearby S1 images are available. The best-performing results are **bolded** and the second best are underlined.

| Sentinel-2 | Sentinel-1 | Landsat-8/9 | Preproc. | PSNR (↑) | SSIM (↑) | SAM (↓) | MAE (↓) |
|---|---|---|---|---|---|---|---|
| ✓ | ✓ | | S1: FO | 28.474 | 0.906 | 6.373 | 0.036 |
| ✓ | | | - | 31.725 | 0.920 | 6.084 | 0.026 |
| ✓ | ✓ | | S1: AM | 30.506 | 0.922 | 6.258 | 0.027 |
| ✓ | ✓ | | S1: FZ | **33.107** | **0.930** | **5.719** | **0.022** |
| ✓ | ✓ | ✓ | S1: FO, LS: FO | 30.040 | 0.898 | 6.989 | 0.033 |
| ✓ | ✓ | ✓ | S1: FZ, LS: FO | 31.416 | 0.914 | 6.622 | 0.026 |
| ✓ | ✓ | ✓ | S1: FZ, LS: FZ | 32.522 | 0.923 | 6.233 | 0.024 |

With the new preprocessing strategy for Sentinel-1 data gaps, we revisit the scaling law. As shown in Table 7, the scaling law holds for both preprocessing methods. Additionally, when it comes to full dataset training, the preprocessing methods do not cause significant differences in the results. Notably, the overall results improve when the Sentinel-1 gaps are filled with constant zeros for the small and medium dataset, indicating a potential inductive bias of filling with constant zeros.

**Table 7:** Scaling law of our model on our AllClear datasets with UnCRtainTS as backbone architecture, with gaps being zeros. Preprocessing methods: *FZ* - Fill zeros. The best-performing results are **bolded** and the second best are underlined.

| Fraction of Data | # data point | Preproc. | PSNR (↑) | SSIM (↑) | SAM (↓) | MAE (↓) |
|---|---|---|---|---|---|---|
| 1% | 2,786 | S1: FZ | 32.039 | 0.922 | 6.469 | 0.024 |
| 3.4% | 10,167 | S1: FZ | 33.107 | 0.930 | 5.719 | 0.022 |
| 10% | 27,861 | S1: FZ | 33.163 | 0.929 | 5.606 | 0.023 |
| 100% | 278,613 | S1: FZ | **34.148** | 0.935 | 5.338 | **0.021** |

### 4.6 Experiments on spatial correlation

To investigate the role of spatial correlation in model generalization, we conducted a geographical hold-out experiment. We trained and validated the model for 16 epochs using a modified version of the AllClear dataset that excluded all North American regions of interest (ROIs). The model was evaluated on two distinct test sets: one containing 2,887 North American ROIs (combined from original train, validation, and test splits) and another containing only non-North American ROIs from the original test set. The results shown in Table 8 suggest that training solely on the held-out dataset does not guarantee spatial generalization. This experiment highlights the importance of addressing spatial generalization in future research. Further investigation of the spatial correlation is presented in Appendix C.4

**Table 8:** Performance evaluation on spatially held-out test sets. The model was trained on data excluding North America.

| Test Region | PSNR ($\uparrow$) | SSIM ($\uparrow$) | SAM ($\downarrow$) | MAE ($\downarrow$) |
|---|---|---|---|---|
| Global | 34.872 | 0.945 | 4.961 | 0.018 |
| North America | 31.561 | 0.903 | 7.027 | 0.034 |

## 5 Limitations and Future Work

While AllClear advances the state of cloud removal in satellite imagery, we acknowledge several important limitations. First, the largest limitation of AllClear is the lack of ground truth annotations. The cloud labels are derived from existing cloud masks computed using s2cloudless algorithm (offered by Google Earth Engine), not manually annotated. Second, we are using Google Earth Engine product level-1C, which is not atmospherically corrected. The main reason is for consistency with the previous largest cloud removal dataset and the derived pre-trained models.

For future work, our findings suggest several promising directions: (1) development of hybrid approaches combining algorithmic and manual cloud annotations, (2) investigation of atmospheric correction's impact on cloud removal performance, and (3) extension to other satellite platforms and spatial resolutions.

## 6 Conclusion

This paper has introduced *AllClear*, the most extensive and diverse dataset available for cloud removal research. The larger training set significantly advances state-of-the-art performance. Our dataset also enables stratified evaluation on cloud coverage and land cover, and ablations of the sequence length and sensor type. We hope that future research can build on this benchmark to advance cloud removal, for instance by exploring the dynamics between SAR and multispectral images.

## Acknowledgements

This research was supported by the National Science Foundation under grant IIS-2144117 and IIS-2403015.

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

# A Overview

In this supplementary material we present more information about the dataset (including a datasheet for the dataset) and extensive results that could not fit in the main paper. In Sec. B we present more details about our dataset such as dataset specifications. In Sec. C we present additional ablation studies. In Sec. D we include a datasheet for our dataset, author statement, and hosting, licensing, and maintenance plan.

The data and pre-trained models are publicly available at `https://allclear.cs.cornell.edu`. Our code for accessing the dataset, benchmark result reproduction can be found at `https://github.com/Zhou-Hangyu/allclear`.

The Croissant metadata tool was not used because it does not support the metadata format we used in our dataset. Specifically, we use a hierarchical structure with dictionaries of lists to store the file path and corresponding timestamp for each image within each sample. The Croissant framework currently does not support parsing such a format. We will provide Croissant metadata file once support for this format is available in the future.

# B   Dataset Curation

We define a sample (i.e., an instance) from the AllClear dataset using an ordered pair $<I_1, I_2, I_3, T, M_1, M_2, M_3, M_T, DW, metadata>$. Specifically, there are three input cloudy images $I_1, I_2, I_3$ and a single target image $T$, each of spatial size $\mathbf{R}^{256 \times 256}$. The number of channels is 13 for Sentinel-2, 2 for Sentinel-1, and 11 for Landsat-8/9. We set Sentinel-2 to be the main sensor (i.e., we evaluate models' performance on reconstructing Sentinel-2 images) and use the other satellites as auxiliary ones. The cloud and shadow masks for input $M_1, M_2, M_3$ and target $M_T$ are all the same size as the inputs, with the number of channels being 5. These channels represent the cloud probability, binary cloud mask, and binary shadow mask with dark pixel thresholds of $0.2$, $0.25$, and $0.3$. Notably, the cloud and shadow masks are paired with and derived from Sentinel-2 input images only. The $DW$ indicates the land cover and land use maps derived from Dynamic World (DW) V1 algorithm [Brown et al., 2022], which have the same spatial size and resolution, with nine classes representing water, trees, grass, flooded vegetation, crops, shrub and scrub, built-up areas, bare land, and snow and ice. The $metadata$ includes geolocation (latitude and longitude), sun elevation and azimuth, and timestamps.

For the benchmark dataset, we ensured that every target image have a corresponding land cover map generated by Dynamic World to enable stratified evaluation. After removing instances without corresponding land cover maps, we found that 98 out of 3,796 original test ROIs were disqualified, so we moved them to the training split to maintain benchmark dataset quantity and quality. For benchmark evaluation, we notice that some ROIs can provide over 30 test instances while some ROIs only have single test instance, and thus we decide to sample one instance for each ROI to avoid oversampling, resulting in 3,698 benchmark instances. Future works can include more test instances as an alternative to gain a more comprehensive evaluation on model performance. The statistics of our dataset are based on the final version after these adjustments.

**Table 9:** *AllClear* Specifications

| Specification | Description |
| --- | --- |
| Satellites | Sentinel-1/2, Landsat-8/9 |
| ROIs | 23708 (train, validation, test: 19013, 997, 3698) |
| Periods | 2022.01.01 - 2022.12.31 |
| Spectrum | Covering all useful bands with raw values |
| Cloud | Covering all cloud coverages without filtering |
| Metadata | Latitude, longitude, time-stamp, sun elevation, sun azimuth |
| File Format | Cloud Optimized GeoTIFF (COG) with ZSTD compression |
| # of images | 4354652 |
| # Sentinel-2 images | 2185076 (train, validation, test: 1755206, 90590, 339280) |
| # Sentinel-1 images | 897239 (train, validation, test: 721991, 38500, 136748) |
| # Landsat-8 images | 637341 (train, validation, test: 510876, 26611, 99854) |
| # Landsat-9 images | 634996 (train, validation, test: 508818, 26535, 99643) |

We also provide the dataset assets in Table 10, specifying the bands we collected for each satellite sensors, the cloud and shadow masks, and the metadata. For Landsat-8/9, we use the Tier 1 TOA (top-of-atmosphere) Reflectance collection from the Google Earth Engine. For cloud and shadow masks, we use the binary cloud mask from Channel 2 and the binary shadow mask from Channel 5 by default for all our experiments.

**Table 10:** List of assets available for each instance.

| Data Type | Channels | Wavelength | Description |
|---|---|---|---|
| **Sentinel-2** | B1 | 443.9 nm (S2A) / 442.3 nm (S2B) | Aerosols. |
| | B2 | 496.6 nm (S2A) / 492.1 nm (S2B) | Blue. |
| | B3 | 560 nm (S2A) / 559 nm (S2B) | Green. |
| | B4 | 664.5 nm (S2A) / 665 nm (S2B) | Red. |
| | B5 | 703.9 nm (S2A) / 703.8 nm (S2B) | Red Edge 1. |
| | B6 | 740.2 nm (S2A) / 739.1 nm (S2B) | Red Edge 2. |
| | B7 | 782.5 nm (S2A) / 779.7 nm (S2B) | Red Edge 3. |
| | B8 | 835.1 nm (S2A) / 833 nm (S2B) | NIR. |
| | B8A | 864.8 nm (S2A) / 864 nm (S2B) | Red Edge 4. |
| | B9 | 945 nm (S2A) / 943.2 nm (S2B) | Water vapor. |
| | B10 | 1373.5 nm (S2A) / 1376.9 nm (S2B) | Cirrus. |
| | B11 | 1613.7 nm (S2A) / 1610.4 nm (S2B) | SWIR 1. |
| | B12 | 2202.4 nm (S2A) / 2185.7 nm (S2B) | SWIR 2. |
| **Sentinel-1** | VV | 5.405 GHz | Dual-band cross-polarization, vertical transmit/horizontal receive. |
| | VH | 5.405 GHz | Single co-polarization, vertical transmit/vertical receive. |
| **Landsat-8/9** | B1 | 0.43 - 0.45 μm | Coastal aerosol. |
| | B2 | 0.45 - 0.51 μm | Blue. |
| | B3 | 0.53 - 0.59 μm | Green. |
| | B4 | 0.64 - 0.67 μm | Red. |
| | B5 | 0.85 - 0.88 μm | Near infrared. |
| | B6 | 1.57 - 1.65 μm | Shortwave infrared 1. |
| | B7 | 2.11 - 2.29 μm | Shortwave infrared 2. |
| | B8 | 0.52 - 0.90 μm | Band 8 Panchromatic. |
| | B9 | 1.36 - 1.38 μm | Cirrus. |
| | B10 | 10.60 - 11.19 μm | Thermal infrared 1, resampled from 100m to 30m. |
| | B11 | 11.50 - 12.51 μm | Thermal infrared 2, resampled from 100m to 30m. |
| **Land use** | Label | - | Pixel-wise land cover labels. |
| **Cloud and shadow masks** | Channel 1 | Cloud probability (%) | Derived from s2cloudless product. |
| | Channel 2 | Binary cloud mask | Derived from thresholding cloud probability at 30. |
| | Channel 3 | Binary shadow mask | Threshold for dark pixel set to 0.20. |
| | Channel 4 | Binary shadow mask | Threshold for dark pixel set to 0.25. |
| | Channel 5 | Binary shadow mask | Threshold for dark pixel set to 0.30. |
| **Metadata** | List of attributes | - | Latitude, longitude, sun elevation, sun azimuth, capture timestamp. |

# C  Experiments

## C.1  Correlation between Cloud Removal Quality and Cloud and Shadow Coverage

We illustrate the relationship between qualitative performance and cloud and shadow coverage in Figure 6. From the left to the right columns, we quantify the cloud and shadow mask using (1) average cloud coverage, (2) average shadow mask coverage, (3) consistent cloud coverage, and (4) consistent shadow coverage. Specifically, consistent cloud (shadow) coverage refers to the percentage of pixels in the input images that are always covered by clouds (shadows). This shows a consistent trend where higher cloud coverage correlates with decreased quality of the target images, consistent with previous observations. The strips in the subplots, especially in the left column at x-axis values of 0.33, 0.67, and 1.0, are due to the fact that some images are fully clouded, resulting in more data points in particular positions in those subplots. During shadow mask synthesis, we discard regions of shadow masks that overlap with cloud masks. Thus images with low shadow percentage may have extremely high or extremely low cloud coverage. This explains the high variance of model performance in the low shadow percentage region.

## C.2  Evaluation of cloud removal preprocessing model's contribution for downstream tasks

It is important to evaluate if cloud removal preprocessing can be beneficial for downstream tasks. To address this, we consider the scenarios of having partially cloudy images and want to infer the land use segmentation map. The goal is to see if models trained on AllClear dataset yield the best downstream segmentation result. To this end, we have conducted additional experiments using our AllClear dataset to create a land use segmentation task.

For dataset curation, we built a land use segmentation dataset using the existing AllClear test set. The dataset contains 2000 training and 400 test images, each 256x256 pixels. Each multispectral image is paired with a corresponding land use map (9 classes). For model training, we trained a 2-D UNet model on the paired training dataset until convergence.

We prepared several versions of the test set images:

- UpperBound: Original clear images (ground truth for cloud removal)
- UnCRtainTS output: Based on 3 cloudy images, using the pre-trained model to yield the predicted clear images for downstream tasks.
- 10% AllClear model output
- 100% AllClear model output
- LeastCloudy: Selecting the least cloudy image from the three input images

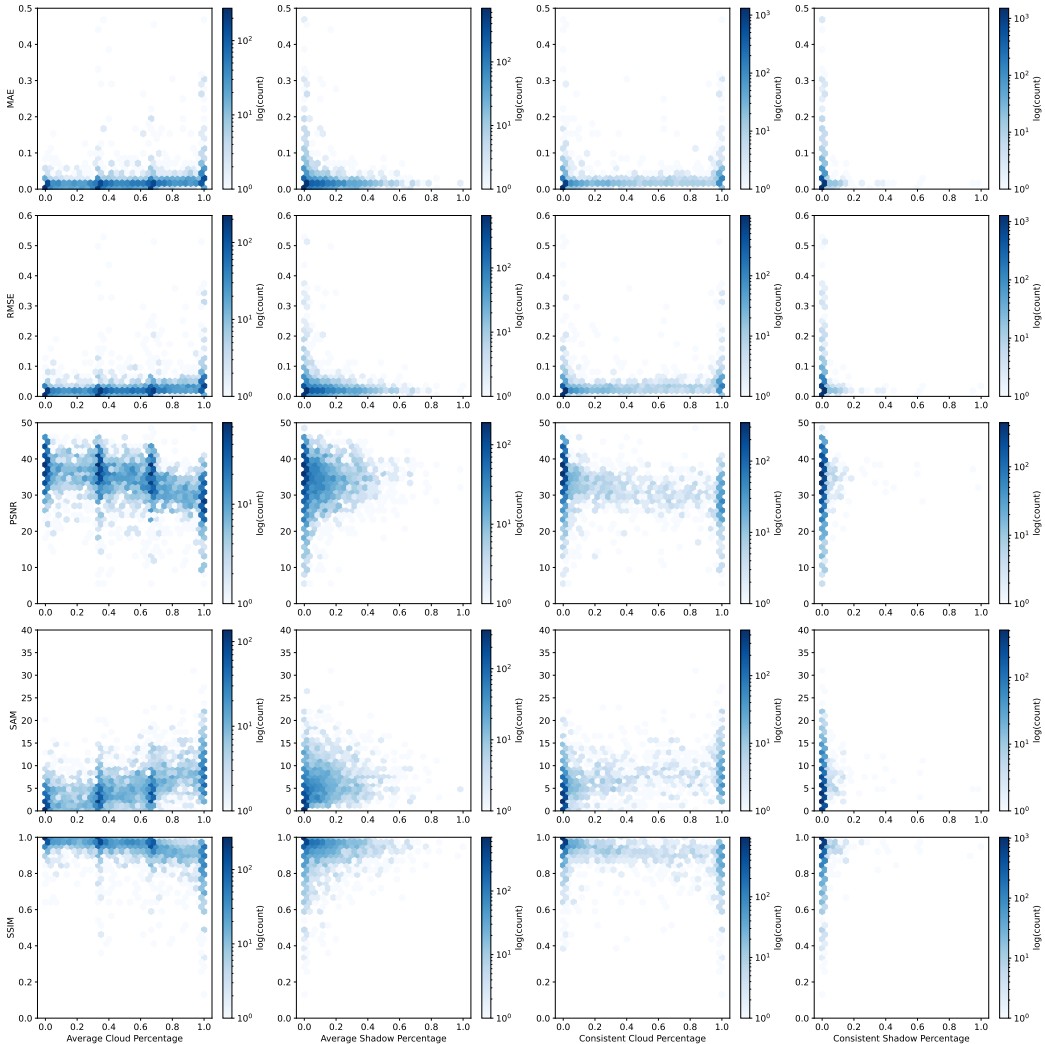

**Figure 6:** Correlation between cloud removal quality and cloud and shadow coverage of UnCRtainTS trained on full AllClear train set, evaluated on the AllClear test set. From left to right, the columns indicate average cloud coverage, average shadow mask coverage, consistent cloud coverage, and consistent shadow coverage. From top to bottom, the rows indicate the metrics MAE, RMSE, PSNR, SAM, and SSIM. The subplots show a consistent trend that a higher cloud coverage rate correlates with lower image reconstruction quality.

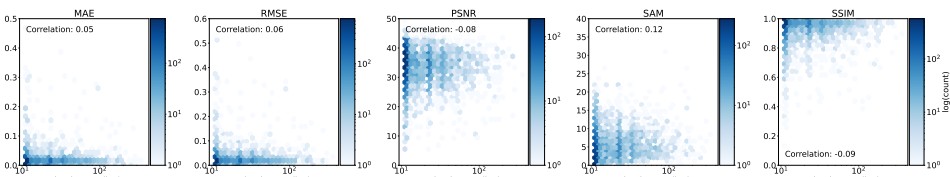

**Figure 7:** Correlation between cloud removal quality and the geodesic distance to the training set for UnCRtainTS trained on 100% of AllClear.

After yielding these different versions of images, we fed them through the trained 2-D UNet and computed relevant metrics. The main objective is to assess whether models trained with AllClear can perform cloud removal in a way that benefits downstream tasks, such as land use segmentation. As shown in Table 11, models trained with 100% AllClear dataset yields the best downstream results, enhancing the Jaccard Index from 0.479 to 0.604, showcasing that our AllClear dataset can be beneficial for downstream applications.

**Table 11:** Model trained on AllClear improves downstream land use segmentation. The best performing values are in **bold** and the second best is underlined.

|  | JaccardIndex | F1Score | Accuracy | Precision | Recall |
|---|---|---|---|---|---|
| UpperBound (GT Test Perf.) | 0.698 | 0.809 | 0.827 | 0.820 | 0.824 |
| UnCRtainTS (original model) | 0.479 | 0.635 | 0.636 | 0.670 | 0.654 |
| LeastCloudy | 0.514 | 0.654 | 0.683 | 0.694 | 0.696 |
| 10% AllClear model | 0.591 | 0.731 | 0.733 | 0.744 | 0.739 |
| 100% AllClear model | **0.604** | **0.744** | **0.757** | **0.754** | **0.757** |

## C.3 Evaluation of reliance on cloud mask

As our source for cloud mask comes from s2cloudless, we discuss the reliance on s2cloudless for evaluation. Specifically, to analyze the impact of imperfect cloud masks on our analysis, we considered two scenarios: (1) False positives: Adding extra masks with jitter noise (uniformly random sampling 10% of pixels, with fixed random seeds), and (2) False negatives: Removing existing cloud masks entirely. Then, we evaluated the similarities between each trained model's performance and the ground truth test images concerning these corrupted masks. This approach simulates the way cloud masks are used to exclude pixels during evaluation.

As shown in Table 12, when cloud masks were corrupted PSNR decreased and SAM increased for all models. However, the decrease in performance is relatively minor, indicating that our evaluation is relatively robust. Additionally, despite the mask corruption, the relative ranking of model performance remained consistent, suggesting that the general trend does not significantly influence comparative performance.

**Table 12:** Evaluation of reliance on cloud mask. With simulated false positive (FP) and false negative (FN) corruption of cloud mask, the performance shows minor decrease.

| Dataset | Corruption | MAE | PSNR | SAM | SSIM |
|---|---|---|---|---|---|
| 10% AllClear | none | 0.023 | 32.997 | 6.038 | 0.923 |
| 10% AllClear | FN | 0.023 | 32.842 | 6.06 | 0.921 |
| 10% AllClear | FP | 0.023 | 32.831 | 6.062 | 0.938 |
| 100% AllClear | none | 0.021 | 33.352 | 5.618 | 0.934 |
| 100% AllClear | FN | 0.021 | 33.19 | 5.641 | 0.95 |
| 100% AllClear | FP | 0.021 | 33.202 | 5.639 | 0.933 |
| SEN12MSCRTS | none | 0.039 | 29.009 | 5.972 | 0.898 |
| SEN12MSCRTS | FN | 0.039 | 28.9 | 5.992 | 0.897 |
| SEN12MSCRTS | FP | 0.039 | 28.891 | 5.995 | 0.905 |

## C.4 Further discussion on spatial correlation

To assess the impact of spatial autocorrelation on cloud removal performance, we compute the correlation between models' performance and the distance of each test ROI to its nearest training ROI. We explore the correlation on the UnCRtainTS model trained on 100% of AllClear on the test set. As shown in Figure 7, we find little correlation between model performance and the distance to training ROIs, suggesting that models do not appear to utilize the spatial autocorrelation nature of satellite images and can generalize to unseen regions.

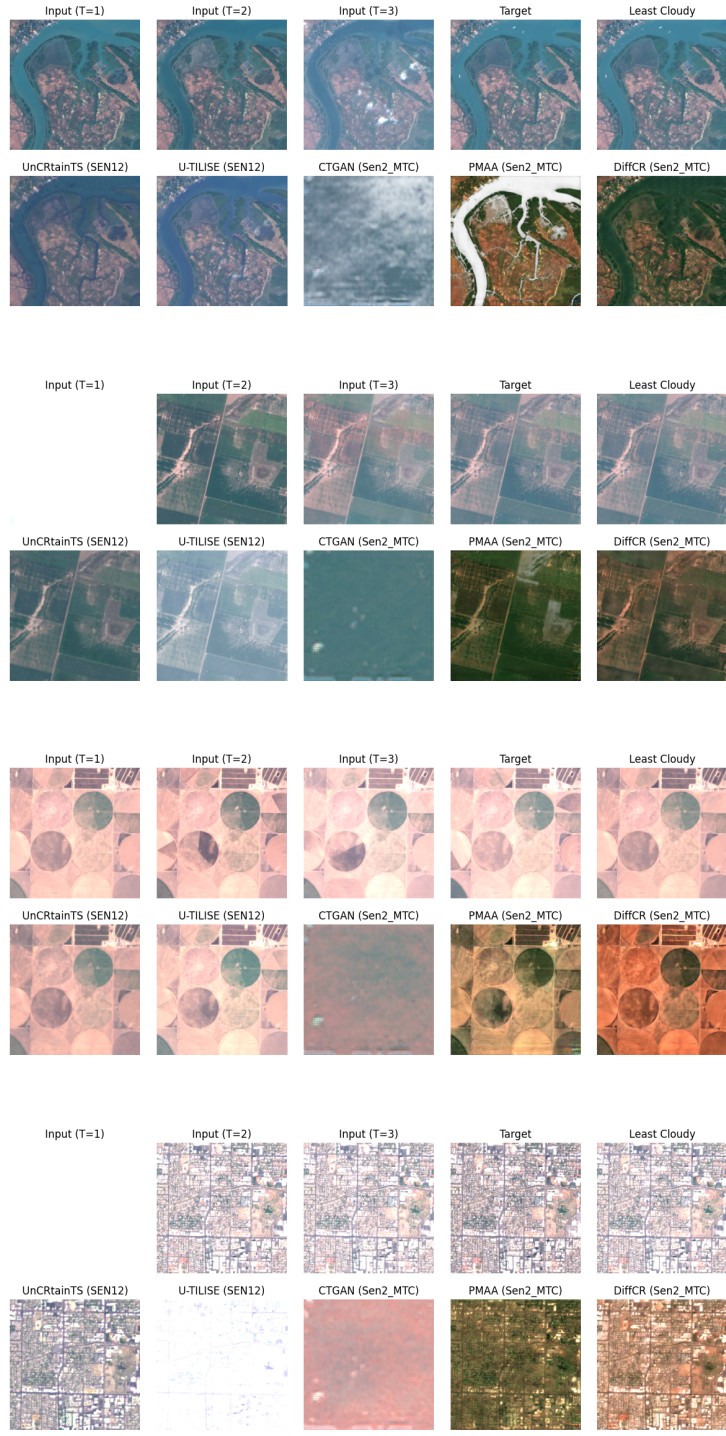

**Figure 8:** Qualitative comparison of the results from different baseline models. The results from four ROIs are shown, including three input images, the target image, the simple baseline result (i.e., Least Cloudy), and the outputs from previous pre-trained models. Specifically, we added the dataset that the model is pre-trained on in the bracket. The results show that the pre-trained UnCRtainTS attains the best qualitative results among all the pre-trained models, while U-TILISE performs well when the input images are mostly clear. On the contrary, CTGAN, PMAA, and DiffCR, pre-trained on a smaller dataset [Huang and Wu, 2022], show several color shifts.

# D   Datasheet

We include a datasheet for our dataset following the methodology from "Datasheets for Datasets" Gebru et al. [2021]. In this section, we include the prompts from  Gebru et al. [2021] in blue, and in black are our answers.

## D.1   Motivation

**For what purpose was the dataset created?** Was there a specific task in mind? Was there a specific gap that needed to be filled? Please provide a description.

The dataset was created to facilitate research development on cloud removal in satellite imagery. The task we include allows a trained model to output a clear image given three (or more) cloudy satellite images. Specifically, our task is more temporally aligned than previous benchmarks.

**Who created the dataset (e.g., which team, research group) and on behalf of which entity (e.g., company, institution, organization)?**

The dataset was created by Hangyu Zhou, Chia-Hsiang Kao, Cheng Perng Phoo, Utkarsh Mall, Bharath Hariharan, and Kavita Bala at Cornell University.

**Who funded the creation of the dataset?** If there is an associated grant, please provide the name of the grantor and the grant name and number.

This work was funded by the National Science Foundation (IIS-2144117 and IIS-2403015).

**Any other comments?**

We specify the bands we collect for Sentinel-1, Sentinel-2, and Landsat-8/9. All images are sampled at 10-meter spatial resolution.

## D.2   Composition

**What do the instances that comprise the dataset represent (e.g., documents, photos, people, countries)?**  Are there multiple types of instances (e.g., movies, users, and ratings; people and interactions between them; nodes and edges)? Please provide a description.

An individual instance in the benchmark dataset is a set of input images, target (clear) images, cloud and shadow masks, land use and land cover maps, and metadata. The input images primarily consist of Sentinel-2 images, while auxiliary sensor information such as Sentinel-1 and Landsat 8/9 may be included if specified in the arguments. Additionally, the number of timestamps for the input images can be 3, 6, or 12, indicating that the inputs contain images from different time frames, typically covering approximately 30 days of image collection, given the average revisit time for Sentinel-2 is 5 days. The cloud and shadow masks are binary spatial maps for each input and target Sentinel-2 image. The land use and land cover maps correspond to the target images. The metadata includes geolocation information such as latitude and longitude, as well as timestamps, sun elevation, sun azimuth, and precomputed cloud coverage.

**How many instances are there in total (of each type, if appropriate)?**

There are 278,613 training instances, 14,215 validation instances, and 55,317 benchmarking instances.

**Does the dataset contain all possible instances or is it a sample (not necessarily random) of instances from a larger set?** If the dataset is a sample, then what is the larger set? Is the sample representative of the larger set (e.g., geographic coverage)? If so, please describe how this representativeness was validated/verified. If it is not representative of the larger set, please describe why not (e.g., to cover a more diverse range of instances, because instances were withheld or unavailable).

The dataset contains all instances from 23,742 ROIs (Regions of Interest) for the year 2022. It does not include all ROIs around the world, but it is a representative subset. We believe the samples are representative of the larger geographic coverage, as the ROI selection was balanced using land use and land cover maps.

**What data does each instance consist of?** "Raw" data (e.g., unprocessed text or images)or features? In either case, please provide a description.

We describe an instance using an ordered pair $<I_1, I_2, I_3, T, M_1, M_2, M_3, M_T, DW, metadata>$. Specifically, there are three input cloudy images $I_1, I_2, I_3$ and a single target image $T$, each of spatial size $\mathbf{R}^{256 \times 256}$. The number of channels is 13 for Sentinel-2, 2 for Sentinel-1, and 11 for Landsat-8/9. The cloud and shadow masks for input $M_1, M_2, M_3$ and target $M_T$ are all the same size as the inputs, with the number of channels being 5. These channels represent the cloud probability, binary cloud mask, and binary shadow mask with dark pixel thresholds of 0.2, 0.25, and 0.3. The $DW$ indicates the land cover and land use maps, which have the same spatial size and resolution, with nine classes representing water, trees, grass, flooded vegetation, crops, shrub and scrub, built-up areas, bare land, and snow and ice. The $metadata$ includes geolocation (latitude and longitude), sun elevation and azimuth, and timestamps.

**Is there a label or target associated with each instance?** If so, please provide a description.

Yes, each instance is paired with a target clear image as ground truth. The target clear images are selected as images with cloud coverage less than $10\%$.

**Is any information missing from individual instances?** If so, please provide a description, explaining why this information is missing (e.g., because it was unavailable). This does not include intentionally removed information, but might include, e.g., redacted text.

All the information is included in the instances.

**Are relationships between individual instances made explicit (e.g., users' movie ratings, social network links)?** If so, please describe how these relationships are made explicit.

Relationships between instances are made explicit in the temporal and spatial domains. Specifically, the metadata for each instance includes information on their corresponding geolocations and timestamps, thereby establishing the relationships between instances based on their location and time of capture.

**Are there recommended data splits (e.g., training, development/validation, testing)?** If so, please provide a description of these splits, explaining the rationale behind them.

We provide a train-validation-test split for our benchmark. The number of instances in train, validation, and test split are 278,613, and 14,215, and 55,317, respectively.

**Are there any errors, sources of noise, or redundancies in the dataset?** If so, please provide a description.

There are no redundancies in the dataset, as each instance is constructed to be non-overlapping with others in the spatiotemporal domain. However, errors in the dataset may arise from the cloud and shadow masks, since the cloud detection module is not yet perfect or $100\%$ accurate, and similarly, the shadow mask may not be entirely accurate as it is derived from the cloud masks.

**Is the dataset self-contained, or does it link to or otherwise rely on external resources (e.g., websites, tweets, other datasets)?** If it links to or relies on external resources, a) are there guarantees that they will exist, and remain constant, over time; b) are there official archival versions of the complete dataset (i.e., including the external resources as they existed at the time the dataset was created); c) are there any restrictions (e.g., licenses, fees) associated with any of the external resources that might apply to a dataset consumer? Please provide descriptions of all external resources and any restrictions associated with them, as well as links or other access points, as appropriate.

The dataset is self-contained as we provide all images with associated masks and metadata. This dataset is free for non-commercial usage and available to the public. For example, using our download code allows for collecting more metadata or other satellite imagery.

**Does the dataset contain data that might be considered confidential (e.g., data that is protected by legal privilege or by doctor–patient confidentiality, data that includes the content of individuals' nonpublic communications)?** If so, please provide a description.

No, Sentinel-1, Sentinel-2, and Landsat-8/9 imageries are free to use for non-commercial usage and publicly accessible.

**Does the dataset contain data that, if viewed directly, might be offensive, insulting, threatening, or might otherwise cause anxiety?** If so, please describe why.

The satellite images have a medium spatial resolution of 10 meters. We do not believe it includes content that is offensive, insulting, or threatening.

**Does the dataset identify any subpopulations (e.g., by age, gender)?** If so, please describe how these subpopulations are identified and provide a description of their respective distributions within the dataset

No, it does not identify any subpopulations.

**Is it possible to identify individuals (i.e., one or more natural persons), either directly or indirectly (i.e., in combination with other data) from the dataset?** If so, please describe how.

No, the images are of medium resolution, making it impractical to identify or track individuals.

**Does the dataset contain data that might be considered sensitive in any way (e.g., data that reveals race or ethnic origins, sexual orientations, religious beliefs, political opinions or union memberships, or locations; financial or health data; biometric or genetic data; forms of government identification, such as social security numbers; criminal history)?** If so, please provide a description.

No, it does not contain sensitive information.

**Any other comments?**

None.

### D.3    Collection Process

**How was the data associated with each instance acquired? Was the data directly observable (e.g., raw text, movie ratings), reported by subjects (e.g., survey responses), or indirectly inferred/derived from other data (e.g., part-of-speech tags, model-based guesses for age or language)?** If the data was reported by subjects or indirectly inferred/derived from other data, was the data validated/verified? If so, please describe how.

The dataset is built upon the publicly available Sentinel-2, Sentinel-1, and Landsat-8/9 satellite imagery.

**What mechanisms or procedures were used to collect the data (e.g., hardware apparatuses or sensors, manual human curation, software programs, software APIs)?** How were these mechanisms or procedures validated?

The raw satellite images were collected using Google Earth Engine APIs [2].

**If the dataset is a sample from a larger set, what was the sampling strategy (e.g., deterministic, probabilistic with specific sampling probabilities)?**

The dataset is not a sample of a larger dataset.

**Who was involved in the data collection process (e.g., students, crowdworkers, contractors) and how were they compensated (e.g., how much were crowdworkers paid)?**

The first authors are involved in the data collection process.

**Over what timeframe was the data collected?** Does this timeframe match the creation timeframe of the data associated with the instances (e.g., recent crawl of old news articles)? If not, please describe the timeframe in which the data associated with the instances was created.

The dataset is built with satellite imagery in the year 2022. The image captured time stamps for each image in each instance are explicitly labeled.

**Were any ethical review processes conducted (e.g., by an institutional review board)?** If so, please provide a description of these review processes, including the outcomes, as well as a link or other access point to any supporting documentation.

---

[2]https://developers.google.com/earth-engine

The study was exempted from IRB as we do not collect any individual/personal information from users.

**Did you collect the data from the individuals in question directly, or obtain it via third parties or other sources (e.g., websites)?**

Our dataset does not contain information about individuals.

**Were the individuals in question notified about the data collection?** If so, please describe (or show with screenshots or other information) how notice was provided, and provide a link or other access point to, or otherwise reproduce, the exact language of the notification itself.

Our dataset does not contain information about individuals.

**Did the individuals in question consent to the collection and use of their data?** If so, please describe (or show with screenshots or other information) how consent was requested and provided, and provide a link or other access point to, or otherwise reproduce, the exact language to which the individuals consented.

Our dataset does not contain information about individuals.

**If consent was obtained, were the consenting individuals provided with a mechanism to revoke their consent in the future or for certain uses?** If so, please provide a description, as well as a link or other access point to the mechanism (if appropriate).

Our dataset does not contain information about individuals.

**Has an analysis of the potential impact of the dataset and its use on data subjects (e.g., a data protection impact analysis) been conducted?** If so, please provide a description of this analysis, including the outcomes, as well as a link or other access point to any supporting documentation.

Our dataset does not contain information about individuals.

**Any other comments?**

None.

### D.4 Preprocessing/cleaning/labeling

**Was any preprocessing/cleaning/labeling of the data done (e.g., discretization or bucketing, tokenization, part-of-speech tagging, SIFT feature extraction, removal of instances, processing of missing values)?** If so, please provide a description. If not, you may skip the remaining questions in this section.

We preprocessed the Sentinel-2 and Landsat-8/9 images with value clipping and normalization. Detailed steps are depicted in Section 3.2.

**Was the "raw" data saved in addition to the preprocessed/cleaned/labeled data (e.g., to support unanticipated future uses)?** If so, please provide a link or other access point to the "raw" data.

We do not do extra pre-processing of the downloaded image dataset. The preprocessing steps are done on the fly.

**Is the software that was used to preprocess/clean/label the data available?** If so, please provide a link or other access point.

Not applicable.

**Any other comments?**

None.

### D.5 Uses

**Has the dataset been used for any tasks already?** If so, please provide a description.

The dataset presented a novel task and has not been used for any tasks yet.

**Is there a repository that links to any or all papers or systems that use the dataset?** If so, please provide a link or other access point.

N/A.

**What (other) tasks could the dataset be used for?**

Our datasets can be used to create benchmarks for sequence-to-sequence cloud removal as well. For example, the input images are a sequence of images where the clear ones are masked, and the target is the original sequence. The provided metadata contains sun position information and capture timestamps, which may be applied for more generative purposes. Our datasets provide a large corpus of cloudy satellite images, which can potentially facilitate developing cloud and shadow detection models.

**Is there anything about the composition of the dataset or the way it was collected and preprocessed/cleaned/labeled that might impact future uses?** For example, is there anything that a dataset consumer might need to know to avoid uses that could result in unfair treatment of individuals or groups (e.g., stereotyping, quality of service issues) or other risks or harms (e.g., legal risks, financial harms)? If so, please provide a description. Is there anything a dataset consumer could do to mitigate these risks or harms?

Our dataset does not contain information about individuals, so it should not result in unfair treatment of individuals or groups.

**Are there tasks for which the dataset should not be used?** If so, please provide a description.

None.

**Any other comments?**

None.

## D.6 Distribution

**Will the dataset be distributed to third parties outside of the entity (e.g., company, institution, organization) on behalf of which the dataset was created?** If so, please provide a description.

Yes, the dataset is publicly available on the internet.

**How will the dataset will be distributed (e.g., tarball on website, API, GitHub)?** Does the dataset have a digital object identifier (DOI)?

The dataset can be downloaded from Cornell's server at `https://allclear.cs.cornell.edu`. The dataset currently does not have a DOI, but we are planning to get one.

**When will the dataset be distributed?**

The dataset is available (since June 2024).

**Will the dataset be distributed under a copyright or other intellectual property (IP) license, and/or under applicable terms of use (ToU)?** If so, please describe this license and/or ToU, and provide a link or other access point to, or otherwise reproduce, any relevant licensing terms or ToU, as well as any fees associated with these restrictions.

The dataset is available under Creative Commons Attribution-NonCommercial 4.0 International License.

**Have any third parties imposed IP-based or other restrictions on the data associated with the instances?** If so, please describe these restrictions, and provide a link or other access point to, or otherwise reproduce, any relevant licensing terms, as well as any fees associated with these restrictions.

Since our dataset is derived from Sentinel-2, Sentinel-1, and Landsat-8/9 images. Please also refer to Sentinel terms of service[3] and Landsat terms of service[4].

---

[3]`https://scihub.copernicus.eu/twiki/do/view/SciHubWebPortal/TermsConditions`
[4]`https://www.usgs.gov/emergency-operations-portal/data-policy`

**Do any export controls or other regulatory restrictions apply to the dataset or to individual instances?** If so, please describe these restrictions, and provide a link or other access point to, or otherwise reproduce, any supporting documentation.

No, there are no restrictions on the dataset.

**Any other comments?**

None.

## D.7  Maintenance

**Who will be supporting/hosting/maintaining the dataset?**

The dataset is hosted and supported by web servers at Cornell. The CS department at Cornell will be maintaining the dataset.

**How can the owner/curator/manager of the dataset be contacted (e.g., email address)?**

Hangyu and Chia-Hsiang can be contacted via email (hz477@cornell.edu, and ck696@cornell.edu). More updated information can be found on the dataset webpage.

**Is there an erratum?** If so, please provide a link or other access point.

No.

**Will the dataset be updated (e.g., to correct labeling errors, add new instances, delete instances)?** If so, please describe how often, by whom, and how updates will be communicated to dataset consumers (e.g., mailing list, GitHub)?

The updates to the dataset will be posted on the dataset webpage.

**If the dataset relates to people, are there applicable limits on the retention of the data associated with the instances (e.g., were the individuals in question told that their data would be retained for a fixed period of time and then deleted)?** If so, please describe these limits and explain how they will be enforced.

Our dataset does not contain information about individuals.

**Will older versions of the dataset continue to be supported/hosted/maintained?** If so, please describe how. If not, please describe how its obsolescence will be communicated to dataset consumers

In case of updates, we plan to keep the older version of the dataset on the webpage.

**If others want to extend/augment/build on/contribute to the dataset, is there a mechanism for them to do so?** If so, please provide a description. Will these contributions be validated/verified? If so, please describe how. If not, why not? Is there a process for communicating/distributing these contributions to dataset consumers? If so, please provide a description.

We also provide the script downloading code in our codebase, which details our downloading configuration to ensure the dataset can be extended and augmented freely without inconsistency. Others may also do so by contacting the original authors about incorporating more fixes/extensions.

**Any other comments?**

None.

## D.8  Author Statement

The authors assume full responsibility for any potential rights violations and the verification of data licensing.

## D.9  Hosting, Licensing, and Maintenance Plan

The benchmarking dataset is hosted on a Cornell server and is licensed under the Creative Commons Attribution-NonCommercial 4.0 International License. The first authors are responsible for maintaining the dataset.

