# OpenReview forum: "AllClear: A Comprehensive Dataset and Benchmark for Cloud Removal in Satellite Imagery"
_NeurIPS.cc/2024/Datasets_and_Benchmarks_Track — NeurIPS 2024 Track Datasets and Benchmarks Poster_

### Official Review · Reviewer_W34B · 2024-07-13
**AllClear**

**Rating:** 7
**Confidence:** 3

**Review:**

The work presented in this paper demonstrates high quality, clarity, originality, and significance in addressing the challenge of cloud removal in satellite imagery.

Pros:
    - Large-scale and diverse dataset: "AllClear" includes a significant number of ROIs with diverse land-use patterns, providing a comprehensive training set for cloud removal techniques.
    - Comprehensive metadata: In addition to satellite imagery, the dataset includes rich metadata such as geolocation, timestamp, land cover maps, and cloud masks, enabling advanced analysis and stratified evaluations.
    - Well-defined benchmarking task: The authors have carefully designed a sequence-to-point benchmarking task, addressing temporal misalignment issues and providing multiple options for cloud and shadow masks.
    - Experimental evaluation: The paper includes experimental results and comparisons with previous state-of-the-art models, demonstrating the effectiveness and advantages of using the "AllClear" dataset.

Cons:
    - Limited discussion on real-world impact: While the paper mentions potential applications, a more detailed discussion on how the dataset can be utilized in real-world scenarios, and the potential benefits to various industries, would be beneficial.
    - Complexity of data preprocessing: The data preprocessing steps, including cloud and shadow mask computation, are intricate and may require significant computational resources, which could be a challenge for researchers with limited access to high-performance computing infrastructure.

**Strengths:**

The "AllClear" dataset exhibits several strengths:
    - Significance: Cloud removal is a crucial task for satellite imagery analysis, with applications in environmental monitoring, disaster response, and urban planning. The large-scale and diverse nature of "AllClear" addresses a significant challenge in this field.
    - Relevance: The dataset is highly relevant to the remote sensing community and researchers working on computer vision tasks related to satellite imagery analysis.
    - Quality: The dataset is carefully constructed, ensuring sufficient coverage of the planet's diversity and including data from multiple satellite sensors. The inclusion of comprehensive metadata enhances its quality and usefulness for research and development.

**Additional Feedback:**

Overall, the "AllClear" dataset is a valuable contribution to the field of satellite imagery analysis, specifically addressing the challenge of cloud removal. The large-scale and diverse nature of the dataset, along with comprehensive metadata, makes it a significant resource for researchers and practitioners. Further exploration of real-world applications and potential collaborations could enhance the impact of this work.

**Clarity:**

The paper is generally well-written and clear. However, some sections, particularly those describing the technical details and data preprocessing steps, could be simplified to improve readability for a broader audience.

**Correctness:**

The claims made in the paper appear correct, and the dataset is constructed in a sound and systematic manner. The experimental results and comparisons provided in the paper support the effectiveness of the "AllClear" dataset for cloud removal tasks.

**Documentation:**

Documentation is sufficient, but Kaggle notebooks with demos could increase research availability.

**Ethics:**

No  ethical concerns.

**Limitations:**

The authors have addressed some limitations, such as the impact of cloud coverage on evaluation metrics and the potential for noise in ground-truth images.
While the dataset focuses on cloud removal, it would be beneficial to evaluate the performance of models trained on "AllClear" when applied to other related tasks or different satellite sensors.

**Opportunities For Improvement:**

While the "AllClear" dataset is a valuable contribution, there are some limitations and opportunities for further enhancement:
    - Task specificity: The dataset is designed for a specific task of cloud removal, and it may not directly translate to other satellite imagery analysis tasks. Expanding the dataset to include other relevant challenges, such as cloud detection or land cover classification, could increase its applicability.
    - Data accessibility: The preprocessing and handling of large-scale satellite imagery may pose challenges for researchers with limited computational resources. Providing preprocessed data or simplified access methods could improve the dataset's accessibility.

**Relation To Prior Work:**

The paper provides a clear differentiation from previous contributions. It highlights the limitations of existing datasets, such as limited geographical diversity and the exclusion of very cloudy images, and explains how "AllClear" addresses these issues.

**Summary And Contributions:**

This paper presents the "AllClear" dataset, which aims to address the challenge of cloud removal in satellite imagery. The contribution lies in introducing the largest public dataset specifically designed for cloud removal, featuring diverse land-use patterns and comprehensive metadata. The dataset includes globally distributed regions of interest (ROIs) with multi-spectral optical imagery, synthetic aperture radar (SAR) imagery, and auxiliary remote sensing products. By providing a large-scale and diverse training set, the authors aim to improve the performance of cloud removal techniques, enabling better environmental monitoring, disaster response, and other applications.

---

> ### Author Rebuttal · Authors · 2024-08-17
>
> **1. Task specificity**
>
> We believe the AllClear dataset is ready for wider applicability for it contains several modalities, including Sentinel-1, Sentinel-2, Landsat imagery, cloud masks, and land use and land cover maps. This comprehensive collection enables the dataset to support various remote sensing tasks, such as cloud detection, Sentinel-1 to Sentinel-2 image translation, Landsat to Sentinel-2 image translation, land use segmentation with Sentinel-1/2 or Landsat, learning a common feature representation for Sentinel-1, Sentinel-2, and Landsat imagery while addressing the clouding problem.
>
> **2. Data accessibility**
>
> We appreciate the reviewer's concern about data accessibility. To make AllClear as useful for more people as possible, we decided to store the raw data as is and perform preprocessing on-the-fly, so that users are not restricted to our preprocessing method and can choose the optimal pipeline for their use cases. As for the preprocessing step, which consists of clipping, normalization, and turning nan to number, it takes 1.02 milliseconds on average to process one Sentinel-2 multispectral image using CPU. The cloud and shadow mask computation is also performed on Google Earth Engine's remote server, requiring little to no computation from AllClear users.
>
> We also have taken several steps to facilitate our users for accessing AllClear with ease:
> 1. We provide a simplified way of accessing the data through a PyTorch data loader in our codebase. The dataset is inherited from PyTorch Dataset and is defined in our code (https://github.com/Zhou-Hangyu/allclear/blob/main/allclear/dataset.py#L63). To initialize the AllClear Dataloader, users can directly apply the PyTorch DataLoader function and adjust parameters such as the number of workers, prefetch numbers, and other settings to optimize loading speed for their specific hardware.
> 2. Customization is also very easy. As shown in our code (https://github.com/Zhou-Hangyu/allclear/blob/main/allclear/benchmark.py#L276), users only need to input a few parameters to customize their data access: ROI list, main sensor selection (typically Sentinel-2), auxiliary sensor selection (such as Sentinel-1, Landsat-1/2), number of input time frames.
>
> Lastly, we acknowledge that the sheer amount of data AllClear offers inevitably requires a large computation workload to process. We created several AllClear subsets with various sizes (from 1%, 3.4%, to 10%) and will package and distribute these along with the full dataset upon acceptance. We hope these subsets can encourage researchers with limited computational resources to partake the cloud removal challenge and make ever better tools together.
>
> **3. While the dataset focuses on cloud removal, it would be beneficial to evaluate the performance of models trained on "AllClear" when applied to other related tasks or different satellite sensors.**
>
> We agree with the reviewer on the importance of evaluating cloud removal preprocessing model’s contribution for downstream tasks. To address this, we consider the scenarios of having partially cloudy images and want to infer the land use segmentation map. The goal is to see if models trained on AllClear dataset yield the best segmentation result. We have conducted additional experiments using our AllClear dataset to create a land use segmentation task. Here are the details of the evaluation.
>
> For dataset curation, we built a land use segmentation dataset using the existing AllClear dataset. The dataset contains 2000 training and 400 test images, each 256x256 pixels. Each multispectral image is paired with a corresponding land use map (9 classes). For model training, we trained a 2-D UNet model (M) on the paired training dataset until convergence.
>
> We prepared several versions of the test set images:
> 1. UpperBound: Original clear images (ground truth for cloud removal)
> 2. UnCRtainTS output: Based on 3 cloudy images, using the pre-trained model to yield the predicted clear images for downstream tasks.
> 3. 10% AllClear model output
> 4. 100% AllClear model output
> 5. LeastCloudy: Selecting the least cloudy image from the three input images
>
> After yielding these different versions of images, we fed them through the trained 2-D UNet and computed relevant metrics. The main objective is to assess whether models trained with AllClear can perform cloud removal in a way that benefits downstream tasks, such as land use segmentation.
> As shown in Table 2 in the rebuttal pdf, models trained with 100% AllClear dataset yields the best downstream results, enhancing the Jaccard Index from 0.479 to 0.604, showcasing that our AllClear dataset can be beneficial for downstream applications.

---

### Official Review · Reviewer_ypDk · 2024-07-22
**Review- Cloud Removal Dataset and Benchmark**

**Rating:** 3
**Confidence:** 5
**Clarity:** Yes the writing is clear

**Review:**

The authors compile a dataset based on several public satellite sensors and existing cloud coverage and shadow algorithms.  They demonstrate that past cloud detection algorithms can be improved by training the model against more data.  While this dataset is large, it is compiled rather than collected- the authors are not providing any new information that is not publicly available.  Curating has some value, but in my opinion, there is not substantial value that this adds to the broader community as practitioners are able to pull together similar datasets- very often the bottleneck is not around creating a large dataset, but around the training cost of producing the model from that.

To that end, it has been demonstrated by many publications (often as a necessary step to some primary analysis on another task) that the existing algorithmic cloud detection algorithms could be augmented with ML models over top.  Again, pulling together a massive dataset isn't the challenge, it's the time and cost to train so something like a large pretrained model may have more benefit to the community as opposed to the dataset itself in this case.  The challenge there is that the impact of the normalization selections the authors made, which would dictate what normalization schema is used for the downstream task, would need to be explored.

The authors have demonstrated that scaling the amount of data dramatically improves result and it appears that even more data would improve results further.  The authors do not state that the current dataset can be readily extended to new ROIs or years but running a set of code, for example.  However, given that the data is not new and the labels are algorithmic, the ability to update/extend the dataset seems like a critical piece that is missing.  There may still need to be manual inspection after the fact, which is fine, but presumably since all the pieces that went into the creation of this dataset were algorithmic, extending it algorithmically seems like a very important, missing piece.

**Strengths:**

The dataset is large and broad with inputs from three different sensors (sentinel-1,2 and landsat 8/9) across many ROIs.

Demonstrating the impact of the performance based on the amount of data is expected, but the magnitude of improvement is good to quantify.

**Additional Feedback:**

Fairly significant preprocessing is done on the data including resampling and clipping.  (How) does this limit the application of this dataset to others who may feel different preprocessing is appropriate to their use case/analysis.

It's unclear form the opening of the paper how the cloud masks are being determined in this dataset- I think this needs to be mentioned earlier than section 3.2.

Results suggest that even more data would continue to improve results.  Can this dataset be extended algorithmically to other ROIs or years?

**Correctness:**

Training protocol including the equipment used, time for training, etc, is all absent.  They state in the checklist that this information is NOT provided, which is a major omission.

**Documentation:**

There is some conflict within the checklist in terms of what is being released (and where).

**Ethics:**

No ethical concerns

**Limitations:**

No societal concerns.

Limitations are not discussed and the authors state in the checklist that this is not provided.  This is a major oversite.

I do think the authors need to speak to the cost of training a model from this dataset as I see this as a clear barrier to usage among the community.

**Opportunities For Improvement:**

The authors are curating rather than collecting the data.  That's fine, but it's not a major contribution as pulling a year of data from these public satellite providers is reasonably straightforward for practitioners in the field.  Similarly they are not improving upon the annotation process is as it leverages an existing algorithm to determine the cloud coverage. See further comments in "Review"

It's not entirely clear how useful this dataset would be in practice.  Users may not always want to use Sentinel 1 and 2 imagery for their downstream task.  So would they go back and pull just Sentinel 1, for example, and then train a model on that.  Related to the comments in review, this still puts a huge computational burden on the end user.  Past models are largely undertrained because of limitations around the cost of the computation, not the ability to create a larger dataset.

Failure cases around changing the color is a major problem for more environmental and agronomic use cases.  Flagging such errors  (e.g. a post processing filter/model) would be very important.

It's unclear what is being released.  The checklist says the data and code are proprietary and will not be released.  If that's true it's not actually a dataset paper.  The fact that more data improves model performance is not a sufficient result for a benchmarks paper (nor really any publication), especially when training details are not provided.

**Relation To Prior Work:**

Prior works are reasonably discussed

**Summary And Contributions:**

The authors introduce a large dataset of almost 24,000 ROIs and 4million images for cloud removal from public satellite.  Data is a year of Sentinel-2 + Landsat 8/9, SAR, cloud masks and land cover maps.  They demonstrate that the size of the dataset (an increase of 30x) increases performance substantialy.

---

> ### Author Rebuttal · Authors · 2024-08-17
>
> **1. Is the dataset and code publicly available?**
>
> Yes, as we stated in Supplementary Material line 8, "The data is publicly available at https://allclear.cs.cornell.edu. Our code for accessing the dataset and benchmark result reproduction can be found at https://github.com/Zhou-Hangyu/allclear." We apologize for the confusion: we accidentally left the example checklist in the submission before the actual checklist. We will remove this example checklist and clarify that data and code are available.
>
> **2. Contribution to data collection.**
>
> We believe our work provides values beyond simple compilation, which addresses several challenges in the field:
> - **Benchmark Creation and Exposing Overlooked Issues**: Our large test set enables detailed stratified evaluation across diverse landscapes and cloud conditions. Our results reveal critical areas for improvement, such as poor performance on water, snow, and ice, pointing to new research directions for the community. These insights were not provided by existing datasets and benchmarks.
> - **Multi-Sensor Synergy**: While users might not always need both Sentinel-1, Sentinel-2, and Landsat-8/9 data, our dataset offers several potential applications. For example, research may be on (a) Sentinel-1 to Sentinel-2 translation tasks, b) exploring how Sentinel-1 can aid in cloud removal for Sentinel-2, or c) learning a common feature representation for Sentinel-1, Sentinel-2, and Landsat imagery while addressing the clouding problem, etc.
> - **Pre-trained Model Release**: We will provide the model pre-trained on the full AllClear dataset on GitHub. Researchers can use lightweight models directly or fine-tune them for specific applications, significantly lowering the barrier to entry for cloud removal research and applications.
>
> **3. Normalization, Sensor Selection, and Extensibility**
>
> - **Clarification on data preprocessing (normalization).** The preprocessing steps mentioned in Section 3.2 (line 143) are not performed during dataset preparation. Instead, it is processed on the fly. _The dataset we provide contains raw values, giving users the freedom to build their models with their preferred preprocessing pipelines._ To avoid confusion, we will move this paragraph to the appendix in the next version with additional clarification. Also, as we aim to establish a consistent benchmark that aligns with existing research, the normalization strategy is consistent with them [1,2].
>
> - **Sensor Selection and Practical Utility.** We chose to focus on Sentinel-1, Sentinel-2, and Landsat imagery for several reasons. While we understand that some users may prefer different sensor combinations, our dataset serves as a comprehensive benchmark for a widely used and researched sensor pair.
>   - These sensors provide the highest spatial resolution imagery freely available.
>   - They represent an active area of research in remote sensing and earth observation [3,4].
>   - They offer a combination of optical and SAR data, which is particularly valuable for cloud removal tasks.
>
> - **Extensibility.** AllClear has been designed with scalability. To facilitate easy extension, we provide dataset downloading code (https://github.com/Zhou-Hangyu/allclear/blob/main/dataset/download_data.py) that allows users to customize various aspects of the data collection process.
>
> **4. Failure cases in color changes (environmental and agronomic use cases)**
>
> We thank the reviewer for noting this. After visualizing the prediction and comparing it with the target data empirically, we found that PSNR, SSIM, and SAM are not a good indicator for color changes. New metrics (perhaps looking at different spatial frequencies) are necessary. That said, our research with AllClear provides some ways of mitigating this concern:
>
> - **Mitigating Color Shift.** As evidenced in Figure 5 of our paper, models trained on the full AllClear dataset show significantly reduced color shifts compared to those trained on smaller subsets (i.e., 10% of AllClear). This demonstrates that a comprehensive dataset is crucial for accurate color preservation.
>
> - **Geographical and Temporal Diversity.** AllClear's global coverage and diverse land use patterns provide the necessary variety to train models to handle real-world scenarios, contributing to more robust color preservation across different environments.
>
> - **Benchmark for Error Detection.** AllClear can readily serve as a robust benchmark dataset for developing and evaluating error detection models. The dataset's comprehensive metadata, including land cover information and cloud/shadow masks, provides a rich context for training such error detection systems.
>
> We believe these aspects of AllClear contribute significantly to addressing color fidelity issues in cloud removal, with potential for further advancements in error flagging and quality assurance crucial for environmental and agricultural use cases.
>
> **5. No Discussion on Limitations and Training Protocol**
>
> Thank you. We have included these discussions in our general response [GR1, GR3].
>
> **6. Cloud Mask Description**
>
> Thank you, we will include them in Introduction.
>
> [1] Stucker, C., Garnot, V. S. F., & Schindler, K. (2023). U-TILISE: A Sequence-to-Sequence Model for Cloud Removal in Optical Satellite Time Series. Remote Sensing.
>
> [2] Ebel, P., Garnot, V. S. F., Schmitt, M., Wegner, J. D., & Zhu, X. X. (2023). UnCRtainTS: Uncertainty quantification for cloud removal in optical satellite time series. CVPR.
>
> [3] Blickensdörfer, L., Schwieder, M., Pflugmacher, D., Nendel, C., Erasmi, S., & Hostert, P. (2022). Mapping of crop types and crop sequences with combined time series of Sentinel-1, Sentinel-2 and Landsat 8 data for Germany. Remote sensing of environment.
>
> [4] Schulz, D., Yin, H., Tischbein, B., Verleysdonk, S., Adamou, R., & Kumar, N. (2021). Land use mapping using Sentinel-1 and Sentinel-2 time series in a heterogeneous landscape in Niger, Sahel. Journal of Photogrammetry and Remote Sensing.

---

### Official Review · Reviewer_jSod · 2024-07-22
**Extensive new cloud removal benchmark dataset with interesting experiments and lack of a discussion section**

**Rating:** 6
**Confidence:** 5
**Correctness:** The paper & dataset appear correct to…
**Clarity:** The paper is written well and easy to…

**Review:**

This paper introduces a new dataset focused on cloud removal in remote sensing imagery. The dataset is well motivated and tackles clear research gaps. The experimental section is informative and should give some interesting insights to practitioners. The paper is currently held back most by the lack of a discussion section, some missing explanations on the experiments and a lack of geographic splits in the dataset. Overall, I believe that this paper can provide a valuable contribution to the ML + EO community if these issues are addressed.

**Strengths:**

The main arguments for the novelty (and necessity) of the dataset are valid. Aligned data from different EO sensors is not often available and relevant for this task. The geographic distribution of EO datasets is another often overlooked factor that this dataset addresses. Lastly, the metadata provided in the dataset can be helpful for determining aspects like performance over space or capture angle. The value of the introduced dataset is somewhat validated by the experiments the authors conduct. But overall, the ablations and different data aspects investigated in the experimental section are most relevant to practitioners. Particularly, I enjoyed the section on "failure cases" in 4.1 and the stratified evaluations in 4.3 (though I believe those should have been expanded).

**Additional Feedback:**

N/A

**Documentation:**

The dataset is packaged up well into a GitHub repository with good documentation.

**Limitations:**

One of my biggest issue with this work is the lack of a thorough discussion and limitations section, though I believe this can be remedied - and the experiments are quite extensive and somewhat address some considerations.

**Opportunities For Improvement:**

My main issue with this paper is the lack of a limitations section that contextualizes advantages and disadvantages as well as challenges and opportunities of the dataset within existing work. The authors should include such a section in a future version of the manuscript. The simple baselines (least cloudy, mosaicking) in the experimental section are not explained - and while I believe I understand these only domain experts will. They should be explained in the main body of the paper.
While the authors evaluate performance over land cover areas, they do not provide spatial splits. Considering the spatial autocorrelation natural to geographic data, such functionalities, or at least an analysis and discussion of spatial effects, should be provided. Lastly, a dataset like AllClear would be great for evaluating recent Earth AI foundation models such as Prithvi and Clay for cloud removal tasks. This should not be too resource intense as these models are publicly available and embeddings from these models can be obtained rather easily. This would be extremely valuable to the community and I would encourage the authors to include such experiments in their final manuscript.

Questions:
- What is the mean minimum distance (in time) between a target image v and the nearest input image u_n? How does that mean minimum distance differ from existing cloud removal benchmark datasets?

**Relation To Prior Work:**

Prior work and how the submissions differs from it is discussed in sufficient detail.

**Summary And Contributions:**

This paper introduces a novel benchmarking dataset for cloud removal in remote sensing imagery. The dataset, AllClear, aims to tackle specific gaps in the existing literature: (1) inclusion of data from different sensors (e.g. optical + radar) and (2) geographic representation (by including data distributed across the globe). The dataset is packaged into a comprehensive GitHub repository. The authors also provide experiments using SotA cloud removal models as baselines. They comment on their findings and discuss failure modes and the performance across e.g. temporal scales and across different land cover types.

---

> ### Author Rebuttal · Authors · 2024-08-17
>
> **1. Lack of Analysis on Spatial Autocorrelation**
>
> Thank you for pointing this out. To assess the impact of spatial autocorrelation on cloud removal performance, we compute the correlation between models’ performance and the distance of each test ROI to its nearest training ROI. We explore the correlation on the UnCRtainTS model trained on 100% of AllClear on the test set (Figure 2 in rebuttal pdf).  We find little correlation between model performance and the distance to training ROIs, suggesting that models do not appear to utilize the spatial autocorrelation nature of satellite images and can generalize to unseen regions. Going further, we will include additional experiments in our camera-ready version where we will hold out one continent as a test set.
>
> **2. Extra experiments with other foundation models**
>
> Indeed, AllClear can serve as a useful benchmark for evaluating other remote-sensing foundation models. We will add experiments in the camera ready where we fine-tune Prithvi for generating cloud-removed images, and use AllClear to benchmark their performance.
>
> **3. Mean Minimum Temporal Difference between Target and the Nearest Input Image**
>
> Thank you for raising this question. For the AllClear training set, our mean minimum temporal difference is 3.83 days with a standard deviation of 1.47 days. And for the test set, the mean is 4.30 days with std being 1.29 days. While previous cloud removal datasets did not reveal such stats in their paper, we randomly sampled 1000 data points from SEN12-MS-CR-TS dataset. And their mean is 124.11 days with a standard deviation of 104.75 days.
>
> **4. Lack of Limitation Section and Simple Baseline Description**
>
> We have addressed these concerns in our general response [GR1] and [GR2]. Thank you.

---

> > ### Comment · Reviewer_jSod · 2024-08-26
> > **Thanks for the reply**
> >
> > Thanks for the replies, no more questions at this point.

---

### Official Review · Reviewer_NUUv · 2024-07-25
**A dataset for cloud removal from remote sensing images**

**Rating:** 8
**Confidence:** 4
**Correctness:** The claims and experiment design seem…

**Review:**

This work is technically sound. The authors address the task of generating a large dataset tailored for cloud removal. The pros are listed in the strengths section below, and the cons are listed in the opportunities for improvements and limitations.

**Strengths:**

The strengths of this paper are: 1) diverse land cover types taken into consideration and utilized to perform a stratified benchmark; 2) data from three sensors are integrated (S1, S2, L8/9).

**Additional Feedback:**

All comments and suggestions were given above.

**Clarity:**

The paper is well written. Below, I list a few comments on possible improvements on clarity:

- the terms "Sen_MTC" seem to be used interchangeably "Sen2_MTC"

L112:  By excluding ocean areas using the GeoPandas package
	- but based on what data?

L130: and Landsat 8/9 [Williams et al., 2006].
	- S1 and S2 data descriptions mention the products utilized, but it is missing for Landsat 8/9.

L158: and projecting cloud regions
	- unclear how this is achieved.

L186: We adopt the following metrics common in cloud removal literature
	- despite being common in the literature, I think they should be cited.

L206: We first notice that simple baselines least cloudy and mosaicing perform well on the dataset
	- these are mentioned in results, but were not introduced before, and they should be.

Table 2
	- I think the full table with the comparisons (Supp. Table 3) should be included and discussed in the main doc.

L230 given its good performance on prior benchmarks.
	- cite

L239: using random sampling from the training dataset while evaluating on the same validation set
	- how is data leakage avoided?

L241: We find that more training data clearly improves accuracy significantly across all metrics, resulting in a more than 10% improvement in PSNR
L245: AllClear’s large training set is extremely useful to help the models fit the data better.
	- "extremely useful" sounds a bit out of proportion, considering that a 10-fold increase in the input dataset only lead to a 10% increase in PSNR

**Documentation:**

The main document and the supplemental material include an adequate detail to insure reproducibility. Links are provided to the code base and the generated dataset, including their licenses. Potential negative societal impacts not discussed, but I don't find any either.

**Limitations:**

The authors do not identify/discuss the limitations nor the potential negative societal impacts of their work. Although, I don't see any potential negative societal impacts. However, there are some important limitations which I think should be discussed: 1) there is no ground-truthed data, the cloud labels are taken from existing cloud masks that were computationally derived (not manually annotated); 2) the dependance and sensitivity to that cloud mask dataset is not assessed nor discussed; and 3) there seems to be little to no improvement by using AllClear in comparison with simpler mosaicking approaches.

**Opportunities For Improvement:**

The authors do not list and discuss the limitations of their work, which is an opportunity for improvement in itself. Below, I list specific lines where some improvement could be made:

L10: We validate the effectiveness of our dataset by benchmarking performance [...] and conducting ablation studies on the temporal length and the importance of individual modalities.
	- I think the ablation experiment is insightful and it would valuable to include those results in the main document and discuss them.

Section 3.2:
	- A more detailed reliance on existing cloud masks (s2cloudless) should be dicussed, including how their accuracy would affect the performance of AllClear.

L214: The good performance of least cloudy and mosaicing is intriguing. We conjecture that part of the reason may be that in AllClear, the temporal gap between input images and target images is smaller, so simply averaging or choosing from the input images is likely to yield good results
	- The fact there is no or little improvement upon simple methods such as mosaicking by selecting the least cloudly or averaging makes me question the significance AllClear.

L414: Did you include the total amount of compute and the type of resources used (e.g., type of GPUs, internal cluster, or cloud provider)? [No]
	- These could be described.

**Relation To Prior Work:**

Yes, this work is well contextualized.

**Summary And Contributions:**

This work presents a dataset for research on cloud removal from optical satellite data. This task is pertains to generating a new image where clouded pixels are replaced with cloud-free pixels; it is not the same as cloud detection/masking.

---

> ### Author Rebuttal · Authors · 2024-08-17
>
> **1. Detailed Reliance on s2cloudless**
>
> We endeavor to discuss the reliance on s2cloudless for evaluation from two aspects and will include the discussion in the updated version of the manuscript.
> - **Qualitative examination of the cloud masks.** We sampled 10 test images and their corresponding cloud masks from our benchmarking dataset. As shown in Figure 1 of the rebuttal pdf, visual inspection revealed that most masks accurately fit the clouds and shadows present in the RGB images.
> - **Quantitative analysis.**
>   - Method: To analyze the impact of imperfect cloud masks on our analysis, we considered two scenarios: (1) False positives: Adding extra masks with jitter noise (uniformly random sampling 10% of pixels, with fixed random seeds), and (2) False negatives: Removing existing cloud masks entirely. Then, we evaluated the similarities between each trained model's performance and the ground truth test images concerning these corrupted masks. This approach simulates the way cloud masks are used to exclude pixels during evaluation.
>    - Results: As shown in Table 1 of the rebuttal pdf, when cloud masks were corrupted PSNR decreased and SAM increased for all models. However, the decrease in performance is relatively minor, indicating that our evaluation is relatively robust. Additionally, despite the mask corruption, the relative ranking of model performance remained consistent, suggesting that the general trend does not significantly influence comparative performance.
>
> **2. Clarification on Model Performance Between Simple Baselines and Models**
>
> We would like to clarify an important point regarding the results presented in Table 2. These results show the performance of models pre-trained on the previous dataset (SEN12MS-CR-TS) when evaluated on our test AllClear dataset. They do not represent the performance of models trained on our full AllClear dataset. The fact that the old pre-trained model performs comparably with baselines necessitates a larger and more diverse dataset for improved cloud removal results. The experiment results in Table 4 indeed show that when trained on our full AllClear dataset, our model achieves significantly better performance:
> - PSNR: 33.87 (compared to 29.82 for the LeastCloudy baseline)
> - SSIM: 0.936 (compared to 0.754 for the LeastCloudy baseline).
>
> We will rearrange the result tables in our camera-ready version to make this comparison more clear to readers.
>
> **3. Move Ablation Experiments to the Main Document**
>
> Thank you. We will move the multi-modality ablation studies (Table 4 in Supplementary) to the main document following the ablation on temporal span (in Section 4.4) in our camera-ready version.
>
> **4. Minor fixes**
> - The terms "Sen_MTC" seem to be used interchangeably "Sen2_MTC"?
>    - Yes, we will use “Sen2_MTC” uniformly.
> - L112: By excluding ocean areas using the GeoPandas package - but based on what data?
>    - We use the Natural Earth dataset naturalearth_lowres to avoid the ocean during sampling.
> - L130: and Landsat 8/9 [Williams et al., 2006]. - S1 and S2 data descriptions mention the products utilized, but it is missing for Landsat 8/9.
>    - Thank you, we will add the description for landsat in the camera-ready version.
> - L158: projecting cloud regions - unclear how this is achieved.
>    - We first estimate the direction of shadow using the sun azimuth, and then stretch cloud regions in that direction with a distance set by heuristics.
> - L186: We adopt the following metrics common in cloud removal literature - despite being common in the literature, I think they should be cited.
>    - We will include the citations in the next-version of the manuscript. Thank you.
> - L206: We first notice that simple baselines least cloudy and mosaicing perform well on the dataset - these are mentioned in results, but were not introduced before, and they should be.
>    - As discussed in general response 2, we will include the detailed description for the simple baselines in the next-version of the manuscript. Thank you.
> - Table 2 - I think the full table with the comparisons (Supp. Table 3) should be included and discussed in the main doc.
>    - We will move them (Supp. Table 3 & 4) to the main manuscript. Thank you.
> - L230 given its good performance on prior benchmarks. - cite
>    - We will include the citations in the next-version of the manuscript. Thank you.
> - L239: using random sampling from the training dataset while evaluating on the same validation set - how is data leakage avoided?
>    - The validation ROIs are held out from the training ROIs from the beginning and kept fixed, models are never trained on them.
> - L241: We find that more training data clearly improves accuracy significantly across all metrics, resulting in a more than 10% improvement in PSNR L245: AllClear’s large training set is extremely useful to help the models fit the data better. - "extremely useful" sounds a bit out of proportion, considering that a 10-fold increase in the input dataset only lead to a 10% increase in PSNR
>    - We thank the reviewers for correction on wording. We will remove “extremely” in the next version paper.

---

> > ### Comment · Reviewer_NUUv · 2024-08-31
> > **Excellent rebuttals, wanted to increase score +2**
> >
> > I am very satisfied with the rebuttals and clarifications the authors have added—or that are warranted to be added to the final version—to my comments as well as to the other reviewers comments, some of which were very crucial for the submission’s quality and relevance. I think this submission was intrinsically already valuable and worth of acceptance, but it lacked several clarifications, which I think have all been well addressed. Therefore, I wanted to update my scores to 8. Unfortunately, it seems neurips closed the option for editing reviews before the published deadline—at least since yesterday (2 days before the deadline) I could no longer see the “edit” button.

---

### Author Rebuttal · Authors · 2024-08-17

We appreciate the reviewers' insightful feedback. Below, we address the main critiques, focusing on adding a limitation discussion section, explaining simple baseline implementations, and detailing computational resource usage. We will incorporate these revisions and corresponding results in the manuscript.

**[GR1] Lack of Limitation Section and Potential Negative Societal Impacts (Reviewer NUUv, jSod, and ypDk)**

We discuss the limitations here and will include them in our manuscript.

1. The largest limitation of AllClear is the lack of ground truth annotations. The cloud labels are derived from existing cloud masks computed using s2cloudless algorithm (offered by Google Earth Engine), not manually annotated;
2. Another limitation is that we are using Google Earth Engine product level-1C, which is not atmospherically corrected. The main reason is for consistency with the previous largest cloud removal dataset [1] and the derived pre-trained models [2].
3. We believe the potential negative societal impact of our work is minimal, since the data source is publicly available through Google Earth Engine and that the 10-meter spatial resolution preserves individual privacy.

**[GR2] No Explanation on Simple Baselines Mechanisms (Reviewer NUUv and jSod)**

We thank the reviewers for your helpful suggestions. And we recognize the lack of a proper introduction to the simple baselines will indeed cause confusion to our readers. Here is an additional section for this information that we will put in our final manuscript, under the “4.1 Benchmarking prior methods on the AllClear test set” section:

**Simple Baselines.** To better contextualize model performance, we follow previous works [1,2] and include two simple baselines: “Least Cloudy” and “Mosaicing”. The former simply uses the input image with the least cloud and shadow coverage as the output. “Mosaicing” operates in the following way: for each image coordinates in the input images if only one image is clear, we directly copy its pixel value; if more than one clear images exist, we take the average of these clear pixel values; if there is no clear image, we fill the gap with 0.5.

**[GR3] Total Amount of Compute and Resource Usage (Reviewer NUUv and ypDk)**

Our training of the UnCRtainTS model on the full (100%) AllClear dataset used 8 A6000 GPUs for 32 hours to train for 20 epochs. We will add these details to the next version of our paper.

[1] Ebel, P., Xu, Y., Schmitt, M., & Zhu, X. X. (2022). SEN12MS-CR-TS: A remote-sensing data set for multimodal multitemporal cloud removal. IEEE Transactions on Geoscience and Remote Sensing.

[2] Ebel, P., Garnot, V. S. F., Schmitt, M., Wegner, J. D., & Zhu, X. X. (2023). UnCRtainTS: Uncertainty quantification for cloud removal in optical satellite time series. CVPR.

---

### Decision · Program_Chairs · 2024-09-26

**Decision:**

Accept (Poster)

**Comment:**

This paper introduces AllClear, a comprehensive dataset and benchmark for cloud removal from satellite imagery. The dataset comprises over 4 million images across 23,742 globally distributed regions of interest (ROIs), sourced from Sentinel-1, Sentinel-2, and Landsat 8/9 sensors. AllClear aims to address the challenge of cloud removal by offering a large, stratified dataset that covers a variety of land-use types, cloud conditions, and temporal spans. The dataset is complemented by metadata, including cloud masks, land cover maps, and geolocation data, which allows for a robust evaluation of state-of-the-art cloud removal techniques.

The average reviewer score for this paper was 6.0, with opinions ranging from 3 to 8. Several reviewers commended the significance of the dataset, particularly its scale, diversity, and the inclusion of multiple sensor modalities. The dataset fills an important gap in the remote sensing literature, where cloud occlusion in satellite imagery has long been a challenge for environmental monitoring, urban planning, and disaster response. One reviewer noted that the dataset's geographic and sensor diversity, along with its potential for use in downstream tasks like land use segmentation, makes it a valuable resource for the Earth observation community.

A key point raised by the reviewers was that the dataset primarily consists of publicly available data, compiled from Sentinel and Landsat satellites. This means that AllClear is not built from newly collected data but rather through the careful curation and integration of existing datasets. However, I personally believe this compilation is far from trivial, as it incorporates multiple modalities (e.g., radar, optical) and provides extensive metadata, making it more robust and comprehensive than many existing datasets in the domain. By offering a unified benchmark for cloud removal, AllClear brings together diverse data sources and makes them accessible for focused research, which in itself is a valuable contribution.

Some reviewers expressed concerns about the dataset's generalizability and the performance of the proposed cloud removal approaches when compared to simpler methods like mosaicking. While the incremental gains of using AllClear-trained models over these baselines could be more pronounced, the authors clarified that AllClear is designed to outperform these methods in complex cloud conditions and across a range of geographies and sensor types. Additionally, they addressed the reliability of the cloud masks used and demonstrated that the dataset remains effective even when the cloud masks are imperfect, which mitigates some of the concerns about potential biases introduced by these masks.

The authors' rebuttal was largely well-received, particularly their responses to concerns about the dataset's utility for other remote sensing tasks. They also highlighted that AllClear-trained models can generalize well to other tasks, such as land use segmentation, and plan to release pre-trained models to ease the computational burden on the research community. The authors also committed to including more ablation studies and making the benchmark's results clearer in the final version of the paper.

In conclusion, while AllClear is composed of existing satellite data, its comprehensive integration of multi-sensor data, rich metadata, and large scale make it a significant contribution to the cloud removal and broader remote sensing community. The dataset's utility for a variety of downstream tasks, as well as its potential to drive further research in cloud removal, justifies its acceptance. Future work should continue to refine its applicability to a wider range of tasks and address computational concerns, but the paper and dataset provide a solid foundation for future advancements in this field.